# Divide and Contrast: Source-free Domain Adaptation via Adaptive Contrastive Learning

Ziyi Zhang[1], Weikai Chen[3], Hui Cheng[2], Zhen Li[4,5], Siyuan Li[6], Liang Lin[2], Guanbin Li[2*]

[1]National Key Laboratory of Novel Software Technology, Nanjing University, Nanjing, China
[2]School of Computer Science and Engineering, Sun Yat-sen University, Guangzhou, China
[3] Tencent America, [4] The Chinese University of Hong Kong, Shenzhen, China
[5] Shenzhen Research Institute of Big Data, Shenzhen, China
[6] AI Lab, School of Engineering, Westlake University, Hangzhou, China
`zhangziyi@lamda.nju.edu.cn, liguanbin@mail.sysu.edu.cn`

## Abstract

We investigate a practical domain adaptation task, called source-free unsupervised domain adaptation (SFUDA), where the source pretrained model is adapted to the target domain without access to the source data. Existing techniques mainly leverage self-supervised pseudo-labeling to achieve class-wise *global* alignment [1] or rely on *local* structure extraction that encourages the feature consistency among neighborhoods [2]. While impressive progress has been made, both lines of methods have their own drawbacks – the "global" approach is sensitive to noisy labels while the "local" counterpart suffers from the source bias. In this paper, we present *Divide and Contrast (DaC)*, a new paradigm for SFUDA that strives to connect the good ends of both worlds while bypassing their limitations. Based on the prediction confidence of the source model, DaC divides the target data into source-like and target-specific samples, where either group of samples is treated with tailored goals under an adaptive contrastive learning framework. Specifically, the source-like samples are utilized for learning *global* class clustering thanks to their relatively clean labels. The more noisy target-specific data are harnessed at the instance level for learning the intrinsic *local* structures. We further align the source-like domain with the target-specific samples using a memory-based maximum mean discrepancy (MMD) loss to reduce the distribution mismatch. Extensive experiments on VisDA, Office-Home, and the more challenging DomainNet have verified the superior performance of DaC over current state-of-the-art approaches. The code is available at `https://github.com/ZyeZhang/DaC.git`.

## 1 Introduction

Deep neural networks have brought impressive advances to the cutting edges of vast machine learning tasks. However, the leap in performance often comes at the cost of large-scale labeled data. To ease the process of laborious data annotation, domain adaptation (DA) provides an attractive option that transfers the knowledge learned from the label-rich source domain to the unlabeled target data. Though most DA approaches require the availability of source data during adaptation, in real-world scenarios, one may only access a source-trained model instead of source data due to privacy issues. Hence, this work studies a more practical task, coded source-free unsupervised domain adaptation (SFUDA), that seeks to adapt a source model to a target domain without source data.

---

*Corresponding author.

36th Conference on Neural Information Processing Systems (NeurIPS 2022).

There are two mainstream strategies to tackle the SFUDA problem. One line of approaches focuses on class-wise *global* adaptation. The key idea is to mitigate domain shift by pseudo-labeling the target data [1, 3] or generating images with target styles [4]. However, either the pseudo images or labels can be noisy due to the domain discrepancy, which would compromise the training procedure and lead to erroneous classifications (Figure 1 (a)). The other direction of research strives to exploit the intrinsic *local* structure [2, 5] by encouraging consistent predictions between neighboring samples. Nonetheless, the closeness of features may be biased by the source hypothesis, which could render false predictions. Further, as it fails to consider the global context, it may generate spurious local clusters that are detrimental to the discriminability of the trained model (Figure 1 (b)).

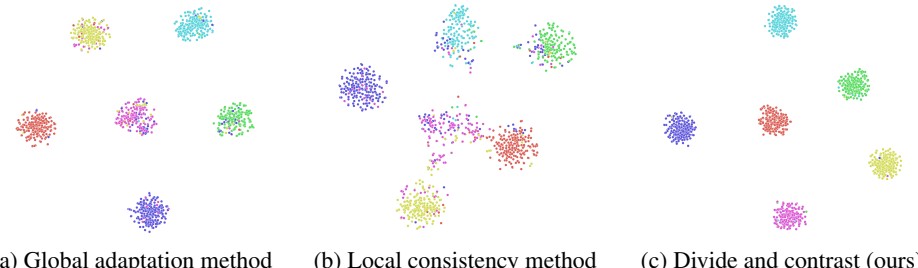

(a) Global adaptation method      (b) Local consistency method      (c) Divide and contrast (ours)

Figure 1: UMAP [6] visualizations of target features trained for 60 epochs on the randomly selected 6 VisDA classes. The results are compared with two types of baselines, global adaption based CPGA [3] and neighborhood consistency based NRC [2]. CPGA can achieve clear global-scale clusters but suffer from false predictions inside each class. NRC can only maintain the intrinsic local consistency but fail to generate clear intra-class boundaries. In contrast, our method inherits the merits of both methods – clear global clusters and strong local consistency, while bypassing their limitations.

To address the above issues, we propose *Divide and Contrast* (*DaC*), a dedicated framework for SFUDA that aims to combine the merits of existing techniques, i.e., taking full advantage of both global and local structures, while sidestepping their limitations. Our key observation is that the "global" approach based on self-training can form clear clusters in the feature landscape (Figure 1 (a)). Its prediction artifacts are mainly caused by noisy labels with low prediction confidence. Hence, after self-training by pseudo labels, we propose to divide the target data into *source-like* and *target-specific* samples, based on the prediction confidence of the source classifier [1]. The two groups of samples are then fully exploited with tailored learning strategies under a unified *adaptive contrastive learning* framework with a memory bank. Specifically, the memory bank consists of representations of all target samples, and momentum updated in the training stage [3, 7]. Thanks to the high prediction confidence, the memory bank generates robust class centroids as positive prototypes for source-like samples. This ensures DaC can obtain a discriminative *global* structure which is robust to noisy pseudo labels. In contrast, for the low-confidence target-specific samples, we ignore their noisy class-level supervisory signals and use the memory bank to generate the positive prototype via the *local* structure. This encourages the network to exploit the intrinsic local structures by contrastive learning.

To prevent the local clustering from forming spurious clusters, as shown in Figure 1 (b), we further transfer the cluster structure from the source-like domain to the target-specific samples. In particular, we propose a maximum mean discrepancy (MMD) based on a memory bank for measuring the distribution mismatch between the source-like and target-specific data. Since the set of source-like and target-specific samples are dynamically updated in our framework, the introduction of a memory bank can effectively alleviate the instability and batch bias caused by mini batches. As seen in Figure 1 (c), the proposed DaC framework can generate clearer and cleaner clusters than both the class-wise and instance discrimination methods. Finally, we provide a rigorous theoretical analysis of the upper bound of the task risk of the target model (Sec. 3) that justifies the soundness of our proposed framework. Extensive experiments demonstrate that our proposed DaC framework has set a new state of the art on a number of challenging datasets, including VisDA, Office-Home, and DomainNet.

We summarize our contributions as follows: 1) a novel divide-and-contrast paradigm for SFUDA that can fully exploit both the global and local structures of target data via data segmentation and customized learning strategies for data subsets; 2) a unified adaptive contrastive learning framework

that achieves class-wise adaptation for source-like samples and local consistency for target-specific data; 3) a memory bank based MMD loss to reduce the batch bias in batch training and an improved MMD loss with non-negative exponential form; 4) new state-of-the-art performance on VisDA, Office-Home, and DomainNet.

## 2   Related Work

**Unsupervised Domain Adaptation.** Conventional UDA methods alleviate the domain discrepancy between the source domain and target domain by leveraging the Maximum Mean Discrepancy (MMD) [8] to match high-order moments of distributions [9, 10, 11] or by adversarial training to learn domain invariant features [12, 13]. In addition, some methods [14, 15] utilize contrastive learning to enhance the global class-level structure by minimizing intra-class distance and maximizing inter-class distance, and others [16, 7] use memory bank to provide class-level information from the source domain and instance-level information from the target domain for contrastive learning. However, the absence of source data makes these methods cannot be applied directly.

**Source-free Unsupervised Domain Adaptation.** Source-free Unsupervised Domain Adaptation (SFUDA) aims to adapt the well-trained source model to the target domain without the annotated source data. Some methods focus on generating target-style images [4] or reconstructing fake source distributions via source model [17]. Another stream of SFUDA methods is exploiting the information provided by the source model. Some methods [1, 18, 19, 3] only leverage the class-level information from the source model and adapt the source model to the target domain by pseudo-labeling, while the other [2, 5] only exploit the neighborhood information and encourage the consistent predictions among samples with highly local affinity.

**Contrastive Learning.** Contrastive learning achieves the promising improvement on unsupervised visual representation learning [20, 21, 22, 23, 24] by learning instance discriminative representations. Although the instance-level contrastive learning has well generalization capability in downstream tasks, it does not perform well on the source-free domain adaptation tasks, which demand correct measurement of inter-class affinities on the unsupervised target domain.

## 3   Preliminaries and Analysis

Our *Divide and Contrast* paradigm mainly divides the target data $\mathcal{D}_T$ into source-like samples $\mathcal{D}_S$ and target-specific outliers $\mathcal{D}_O$ via the source classifier. We claim the consistency robustness (Claim 3.1) of the source-like samples, and further show in Theorem 3.2 an upper bound of task error on the target domain. In this part, we introduce some notations, assumptions, and our theoretical analysis.

**Preliminary**. Only with a well-trained model on source domain, the goal of the SFUDA task is to learn a model $h$ which minimizes the task risk $\epsilon_{\mathcal{D}_T}(h)$ in the target domain, *i.e.* $\epsilon_{\mathcal{D}_T}(h) = \mathbb{P}_{\mathcal{D}_T}[h(x) \neq h^*(x))]$, where $\mathcal{D}_T$ is our available $n_t$ unlabeled *i.i.d* samples from target domain, and $h^*$ is the ideal model in all model space $\mathcal{H}$. Specifically, we consider a $C$-way classification task, where the source and target domain share the same label space. The model $\bar{h}$ consists of a feature extractor $\phi$ and a classifier $g$, *i.e.* $\bar{h}(x) = g(\phi(x))$, which maps input space $\mathbb{R}^I$ to prediction vector space $\mathbb{R}^C$, and $h(x) = \arg\max_c \bar{h}(x)_{[c]}$. The source model is denoted as: $h_s = g_s \circ \phi_s$. The feature from the feature extractor is denoted as $\boldsymbol{f}_i = \phi(x_i)$. For the training stage, the batch data randomly sampled from $\mathcal{D}_T$ is denoted as $\mathcal{B}_T$, and $\delta$ is the softmax function.

**Assumptions**. Following the subpopulation assumption in [25, 26, 27, 28], we also denote $\mathcal{D}_{T_i}$ the conditional distribution of target data $\mathcal{D}_T$ given the ground-truth $y = i$, and further assume $\mathcal{D}_{T_i} \cap \mathcal{D}_{T_j} = \emptyset$ for all $i \neq j$. To introduce the expansion assumption [27, 26], we first define that the suitable set of input transformations $\mathcal{B}(x)$ takes the general form $\mathcal{B}(x) = \{x' : \exists A \in \mathcal{A} \ s.t. \|x' - A(x)\| < r\}$ for a small radius $r > 0$ and a set of data augmentations $\mathcal{A}$. Then, the neighborhood of a sample $x \in \mathcal{D}_{T_i}$ is defined as $\mathcal{N}(x) := \mathcal{D}_{T_i} \cap \{x' | \mathcal{B}(x) \cap \mathcal{B}(x') \neq \emptyset\}$, as well as that of a set $S \subset \mathcal{D}_T$ as: $\mathcal{N}(S) := \cup_{x \in S} \mathcal{N}(x)$. The consistency error of $h$ on domain $\mathcal{D}_T$ is defined as : $\mathcal{R}_{\mathcal{D}_T}(h) = \mathbb{E}_{\mathcal{D}_T}[\mathbb{1}(\exists x' \in \mathcal{B}(x) \ s.t. \ h(x') \neq h(x))]$, which indicates the model stability of local structure and input transformations. To this end, we introduce the expansion assumption to study the target domain.

**Definition 3.1** (($q, \gamma$)-**constant expansion** [27, 26])). *We say Q satisfies ($q, \gamma$)-constant expansion for some constant $q, \gamma \in (0, 1)$, if for any set $S \subset Q$ with $\mathbb{P}_Q[S] > q$, we have $\mathbb{P}_Q[\mathcal{N}(S) \setminus S] > \min\{\gamma, \mathbb{P}_Q[S]\}$.*

**Theoretical analysis**. Our *Divide and Contrast* paradigm mainly divides the target data into source-like and target-specific samples via the source classifier. Specifically, by freezing the source classifier *i.e.* $h = g_s \circ \phi$ [1], we select confident samples with prediction probability greater than a threshold $\tau_c$, and regard them as source-like samples: $\mathcal{D}_S = \{x_i | \max_c \delta(h(x_i)) \geq \tau_c, x_i \in \mathcal{D}_T\}$, and the rest target data is target-specific samples $\mathcal{D}_O = \mathcal{D}_T \setminus \mathcal{D}_S$. Denote by $\mathcal{D}_{S_i}$ the conditional distribution of $\mathcal{D}_S$ where $\mathcal{D}_{S_i} = \mathcal{D}_S \cap \mathcal{D}_{T_i}$. The definition is similar for $\mathcal{D}_{O_i}$. The following claim guarantees the existence of $\tau_c$ and the consistency robustness of source-like samples:

**Claim 3.1.** *Suppose $h$ is $L_h$-Lipschitz w.r.t the $L_1$ distance, there exists threshold $\tau_c \in (0, 1)$ such that the source-like set $\mathcal{D}_S$ is consistency robust, i.e. $\mathcal{R}_{\mathcal{D}_S}(h) = 0$. More specifically,*

$$\tau_c \geq \frac{L_h r}{4} + \frac{1}{2}.$$

**Remark 1.** *Claim 3.1 illustrates the source-like set with a small consistency error, as long as the $\tau_c$ is large enough. Moreover, we empirically claim that model predictions on $\mathcal{D}_S$ are more robust, i.e. $\epsilon_{\mathcal{D}_S}(h) \leq \epsilon_{\mathcal{D}_T}(h)$ [18]. The great properties of the source-like set, consistency, and robustness, motivate us to transfer knowledge from source-like samples to target-specific samples, by contrastive learning and distribution matching.*

Assume we have a pseudo-labeler $h_{pl}$ based on the source model. The following theorem establishes the upper bound on the target risks and states the key idea behind our method. The proofs of both the claim and theorem are provided in Appendix A.

**Theorem 3.2.** *Suppose the condition of Claim 3.1 holds and $\mathcal{D}_T, \mathcal{D}_S$ satisfies ($q, \gamma$)-constant expansion. Then the expected error of model $h \in \mathcal{H}$ on target domain $\mathcal{D}_T$ is bounded,*

$$\epsilon_{\mathcal{D}_T}(h) \leq (\mathbb{P}_{\mathcal{D}_T}[h(x) \neq h_{pl}(x)] - \epsilon_{\mathcal{D}_S}(h_{pl}) + q) \frac{\mathcal{R}_{\mathcal{D}_T}(h)(1 + \gamma)}{\gamma \cdot \min\{q, \gamma\}} + \max_{i \in [C]}\{d_{\mathcal{H} \Delta \mathcal{H}}(\mathcal{D}_{S_i}, \mathcal{D}_{O_i})\} + \lambda,$$

*where constant $\lambda$ w.r.t the expansion constant $q$ and task risk of the optimal model.*

**Remark 2.** *The Theorem 3.2 states that the target risk is bounded by the following three main parts, the fitting accuracy between model $h$ and pseudo-labeler $\mathbb{P}_{\mathcal{D}_T}[h(x) \neq h_{pl}(x)]$, the consistency regularization $\mathcal{R}_{\mathcal{D}_T}(h)$, and the $\mathcal{H}$-divergence between the source-like samples target-specific samples $d_{\mathcal{H} \Delta \mathcal{H}}(\mathcal{D}_{S_i}, \mathcal{D}_{O_i})$. To constrain the above three parts, the theoretical insight of our method lies in class-wise adaptation, consistency of local structure, and alignment between target-specific and source-like samples.*

Based on the theoretical analysis, our method consists of three parts: 1) achieves preliminary class-wise adaptation by $\mathcal{L}_{self}$, which fits $h$ to pseudo-labels; 2) leverages adaptive contrastive loss $\mathcal{L}_{con}$ to jointly achieve robust class-wise adaptation for source-like samples and local consistency regularization for target-specific samples; 3) minimizes the discrepancy between the source-like set and target-specific outliers by $\mathcal{L}_{EMMD}$. The entire loss is defined as:

$$\mathcal{L} = \mathcal{L}_{con} + \alpha \mathcal{L}_{self} + \beta \mathcal{L}_{EMMD}. \tag{1}$$

## 4 Divide and Contrast for Source-free Unsupervised Domain Adaptation

In this section, we introduce *DaC*, our proposed method for source-free unsupervised domain adaptation. As the overview shown in Figure 2, our method consists of three parts, self-training by pseudo-labels, an adaptive contrastive learning framework, and distribution alignment. The overall algorithm of DaC is summarized in Appendix C.

### 4.1 Data Segmentation and Self-Training

To achieve preliminary class-wise adaptation without target annotations, we generate pseudo-labels to supervise the transformation set of input $\mathcal{B}(x)$. In practice, we consider two different augmentation

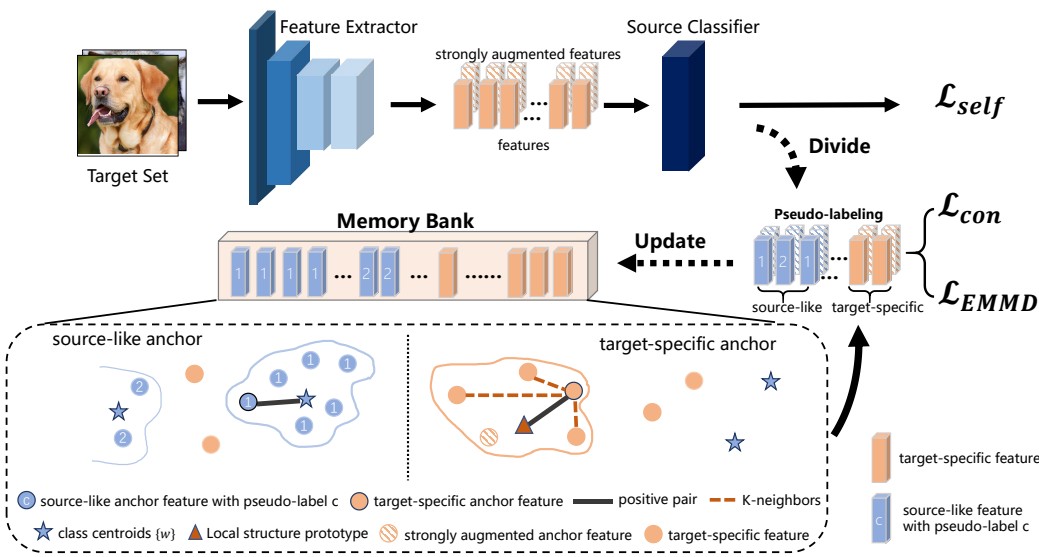

Figure 2: The illustration of the proposed *DaC* framework.

strategies in augmentation set: $\mathcal{A} = \{\mathcal{A}_w, \mathcal{A}_s\}$, where weak augmentation $\mathcal{A}_w$ refers to random cropping and flipping, and strong augmentation $\mathcal{A}_s$ refers to the automatically learned augmentation strategy in [29].

**Pseudo-labeling**. We apply the strategy proposed in [1] to update pseudo-labels in each epoch. Denote by $p_i = \delta(g_s(\boldsymbol{f}_i))$ the prediction from source classifier. The initial centroid for each class $k$ is attained by: $\mathbf{c}_k = \frac{\sum_i^{n_t} \boldsymbol{f}_i p_i[k]}{\sum_i^{n_t} p_i[k]}$. The centroids estimate the centers of different categories, and the samples are labeled identically with its nearest initial centroid:

$$\tilde{\mathcal{Y}} = \{\tilde{y}_i | \tilde{y}_i = \arg\max_k \cos(\phi(x_i), \mathbf{c}_k), x_i \in \mathcal{D}_T\}, \tag{2}$$

where $\cos(\cdot, \cdot)$ is the cosine similarity. The $k$-th centroid is further modified by: $\mathbf{c}_k = \frac{\sum_i^{n_t} \mathbb{1}(\tilde{y}_i = k)\mathbf{f}_i}{\sum_i^{n_t} \mathbb{1}(\tilde{y}_i = k)}$, where $\mathbb{1}(\cdot)$ is the indicator function, and pseudo-labels are further updated by the modified centroids $\mathbf{c}_k$ based on Eqn. 2 in each epoch.

**Self-training for Consistency Regularization**. During the training iteration, we transfer each sample $x_i$ into two views: $\mathcal{A}_w(x_i), \mathcal{A}_s(x_i)$. Network predictions on the two views are denoted as $p_i^w$ and $p_i^s$ respectively. Denote by $\hat{y}_i = \text{onehot}(\tilde{y}_i)$ the one-hot coding of pseudo-labels. Since the same semantic content of the two augmented views, we encourage consistent predictions between two types outputs by the following loss:

$$\mathcal{L}_{self} = -\mathbb{E}_{x_i \in \mathcal{B}_T} \left[ \sum_{c=1}^{C} \hat{y}_i^c \log(p_i^w[c]) + \hat{y}_i^c \log(p_i^s[c]) \right] + \sum_{c=1}^{C} \text{KL}(\bar{p}_c || \frac{1}{C}) + \omega H(p_i^w), \tag{3}$$

where $\bar{p}_c = \mathbb{E}_{\mathcal{B}_T}[p_i^w[c]]$ is regularized by uniform distribution to encourage output diversity, $H(p_i^w) = -\mathbb{E}_{\mathcal{B}_T}[\sum_c p_i^w[c] \log(p_i^w[c])]$ is the Shannon entropy [30], which is used to encourage confident outputs and accelerate convergence, and $\omega$ is the corresponding parameters.

**Dataset Division.** As the analysis before, we divide all samples in $\mathcal{D}_T$ into the source-like set and target-specific outliers. During the training process, samples are dynamically grouped into the source-like set by the threshold $\tau_c$:

$$\tilde{\mathcal{D}}_S^k = \{x_i | \max_c(p_i^w[c]) \geq \tau_c, \arg\max_c(p_i^w[c]) = k, x_i \in \mathcal{D}_T\}, \tag{4}$$

and the target-specific samples are the rest target data, *i.e.* $\tilde{\mathcal{D}}_O = \mathcal{D}_T \setminus \tilde{\mathcal{D}}_S$, where $\tilde{\mathcal{D}}_S = \cup_{k=1}^{C} \tilde{\mathcal{D}}_S^k$. More specifically, $\tilde{\mathcal{D}}_O^k = \{x_i | x_i \in \tilde{\mathcal{D}}_O, \tilde{y}_i = k, \tilde{y}_i \in \tilde{\mathcal{Y}}\}$. To this end, we design an adaptive contrastive learning framework that jointly adapts source-like features to class-wise prototypes and optimizes target-specific features by local structures.

## 4.2 Adaptive Contrastive Learning

Similar to [20, 16], we employ a momentum updated memory bank to store all target features $\mathcal{F} = \{z_i\}_{i=1}^{n_t}$. The memory bank dynamically generates $C$ source-like class centroids $\{w_c\}_{c=1}^C$ and $n_o$ target-specific features $\{v_j\}_{j=1}^{n_o}$, where $n_o = |\tilde{\mathcal{D}}_O|$ is the current number of target-specific samples, which dynamically changes in the training stage.

For a general anchor feature during training time $f_i = \phi(\mathcal{A}_w(x_i))$, $x_i \in \tilde{\mathcal{D}}_S \cup \tilde{\mathcal{D}}_O$, we conduct prototype contrastive loss with similarity measured by inner product:

$$\mathcal{L}_{con} = -\mathbb{E}_{x_i \in \mathcal{B}_T} \log \frac{\exp(f_i \cdot k^+ / \tau)}{\exp(f_i \cdot k^+ / \tau) + \sum_{j=1}^{C+n_o-1} \exp(f_i \cdot k_j^- / \tau)}, \tag{5}$$

where the sum is over one positive pair $(f_i, k^+)$ and $C + n_o - 1$ negative pairs $\{(f_i, k^-)\}$, the temperature $\tau$ is a hyper-parameter and is empirically set as 0.05.

### 4.2.1 Prototype Generation

The contrastive loss tries to classify anchor feature $f$ as its positive prototype $k^+$ from all negative samples. Memory bank generates $C + n_o$ features, including the class-level signals, $i.e.$ $C$ class centroids $\{w\}$ as well as $n_o$ target-specific instance features $\{v\}$. The source-like and target-specific samples are jointly optimized by generating corresponding positive prototypes.

**For the source-like anchor**, $i.e.$ $x_i \in \tilde{\mathcal{D}}_S^k$, since the network prediction $k$ (in Eqn. 4) is relatively reliable [18], we encourage contrastive learning to achieve class-wise adaptation by designating the positive prototype as class centroid $w_k$, $i.e.$ $k^+ = w_k$. The rest $C - 1$ class centroids and $n_o$ target-specific features are used to form negative pairs.

**For the target specific anchor**, due to its noisy class-level supervisory signal, we generate a positive prototype $k^+$ by introducing *Local Structure*, including neighborhood consistency and transformation stability. Specifically, we enhance the semantic consistency with the strongly augmented features $f_i^s = \phi(\mathcal{A}_s(x_i))$ and the $K$-nearest features $\mathcal{N}_K(f_i)$, that can be easily found in memory bank via cosine similarity: $\mathcal{N}_K(f_i) = \{z_j | top\text{-}K(\cos(f_i, z_j)), \forall z_k \in \mathcal{F}\}$. To this end, the positive prototype $k^+$ is generated as: $k^+ = \frac{1}{K+1}\left(f_i^s + \sum_{k=1}^K z_k\right)$, where $z_k \in \mathcal{N}_K(f_i)$. Note that $k^+$ depicts the local information of its corresponding feature $v_i$, and we use the rest $n_o - 1$ target-specific features $\{v_j\}_{j \neq i}$ and $C$ class centroids to form negative pairs.

**Discussion**. For the *source-like anchor*, the positive prototype $k^+ = w_k$ is the mean of all source-like features with confident and consistent predictions, which makes the class-wise adaptation more robust to noisy pseudo-labels. For the $i$-th *target-specific anchor*, its corresponding memory bank feature $z_i \in \mathcal{F}$ can be found in $\mathcal{N}_K(f_i)$. The contrastive loss constrains the consistency error $\mathcal{R}_{\mathcal{D}_T}$ in Thm 3.2. Unlike the previous contrastive method [3], Eqn. 5 jointly achieves class-wise adaptation and instance-wise adaptation. The source-like and target-specific samples are dynamically updated, and adaptively benefit representation learning.

### 4.2.2 Memory Bank

One of the most important reasons to utilize the memory bank is to conserve source information by momentum updating features. Therefore, the memory bank is initialized with the features by performing forward computation of source feature extractor $\phi_s$: $\mathcal{F} = \{z_i | z_i = \phi_s(x_i), x_i \in \mathcal{D}_T\}$. At each iteration, the memory bank features are updated via momentum strategy for the $i$-th input feature: $z_i = mz_i + (1-m)f_i$, where $m \in [0,1]$ is the momentum coefficient and is empirically set as 0.2.

The memory bank provides source-like centroids and target-specific features for contrastive learning. The target-specific features can be directly accessed in memory bank: $\{v_i | v_i = z_i, x_i \in \tilde{\mathcal{D}}_O\}$. We initialize the source-like set before initializing the source-like centroids. Samples with confident predictions from the source classifier are termed source-like. To guarantee sufficient samples in a source-like set in the early training stage, we initialize the source-like set by drawing samples with top 5% predictions in each class. The $c$-th source-like set is initialized as:

$$\tilde{\mathcal{D}}_S^c = \arg\max_{|\mathcal{X}|=N, \hat{\mathcal{X}} \subseteq \mathcal{D}_t} \sum_{x_i \in \mathcal{X}} p_i^w[c], \tag{6}$$

where $N = 5\%n_t$ is the number of samples in each class. And $\tilde{\mathcal{D}}_S^c$ is updated in training stage by Eqn. 4. After updating the source-like set $\tilde{\mathcal{D}}_S (= \bigcup_{c=1}^C \tilde{\mathcal{D}}_S^c)$, the $c$-th class centroids $\boldsymbol{w}_c$ is generated by the mean of all source-like features:

$$\boldsymbol{w}_c = \frac{1}{|\tilde{\mathcal{D}}_S^c|} \sum_{x_i \in \mathcal{D}_S^c} \boldsymbol{z}_i. \tag{7}$$

### 4.3 Distribution Alignment

To obtain better adaptation performance, we further achieve alignment by minimize the $\mathcal{H}$-divergence between source-like set and target-specific samples, $i.e.$ $d_{\mathcal{H}\Delta\mathcal{H}}(\mathcal{D}_{S_i}, \mathcal{D}_{O_i})$, where the kernel-based Max Mean Discrepancy (MMD) [8] is widely used. We use $\tilde{\mathcal{D}}_S^i, \tilde{\mathcal{D}}_O^i$ to estimate $\mathcal{D}_{S_i}, \mathcal{D}_{O_i}$, the related optimization target is: $\min_{h \in \mathcal{H}} d_{MMD}^\kappa(\tilde{\mathcal{D}}_S^i, \tilde{\mathcal{D}}_O^i)$, where $\kappa$ is the kernel function. MMD depicts the domain discrepancy by embedding two distribution into reproducing kernel Hilbert space. Considering the MMD between two domain $\mathcal{S}, \mathcal{T}$ with the linear kernel function in mini-batch (batch size $B$):

$$d_{MMD}(\mathcal{S}, \mathcal{T}) = \frac{1}{m} \sum_{i=1}^m \boldsymbol{s}_i \left( \frac{1}{m} \sum_{i'=1}^m \boldsymbol{s}_{i'} - \frac{1}{n} \sum_{j'=1}^n \boldsymbol{t}_{j'} \right) + \frac{1}{n} \sum_{i=1}^n \boldsymbol{t}_i \left( \frac{1}{n} \sum_{j'=1}^n \boldsymbol{t}_{j'} - \frac{1}{m} \sum_{i'=1}^m \boldsymbol{s}_{i'} \right). \tag{8}$$

where $\boldsymbol{s}, \boldsymbol{t}$ represent the features of two domain, $m, n$ are their corresponding amounts in mini-batch ($m + n = B$). The previous UDA methods regard $\mathcal{S}, \mathcal{T}$ as the source and the target domain, respectively. Two domains take an equal number of samples ( $i.e.$ $m = n = B/2$). Since the absence of source data, we use MMD to measure the discrepancy between source-like and target-specific samples, whose amounts $m, n$ are uncertain in the random batch sampling. The vanilla MMD is not applicable in our setting because samples in mini-batch have estimation bias of the whole distribution, especially when the $m$ or $n$ is small. To this end, we use features in the memory bank to estimate the whole distribution and reduce batch bias. Specifically, for $\mathcal{S} = \tilde{\mathcal{D}}_S^c, \mathcal{T} = \tilde{\mathcal{D}}_O^c$, we replace $\frac{1}{m} \sum_{i'=1}^m \boldsymbol{s}_{i'}$ with $\mathbb{E}_{\tilde{\mathcal{D}}_S^c}[\boldsymbol{z}] = \boldsymbol{w}_c$, $\frac{1}{n} \sum_{j'=1}^n \boldsymbol{t}_{j'}$ with $\mathbb{E}_{\tilde{\mathcal{D}}_O^c}[\boldsymbol{z}]$, and rewrite Eqn. 8 as our linear memory bank-based MMD:

$$\mathcal{L}_{LMMD} = \mathbb{E}_{x_i \in \mathcal{B}_T} \boldsymbol{f}_i(\boldsymbol{q}_i^- - \boldsymbol{q}_i^+), \tag{9}$$

where $\boldsymbol{q}_i^-$ is the correlating prototype in the memory bank at the same domain. For example, if $\boldsymbol{f}_i$ is from the source-like set, $\boldsymbol{q}_i^- = \boldsymbol{w}_c, \boldsymbol{q}_i^+ = \mathbb{E}_{\tilde{\mathcal{D}}_O^c}[\boldsymbol{z}]$. To avoid the negative term in Eq. 9, we improve it as a non-negative form. By simply clipping $\max\{0, \boldsymbol{fq}^- - \boldsymbol{fq}^+\}$, we have:

$$\max\{0, \boldsymbol{fq}^- - \boldsymbol{fq}^+\} = \max\{\boldsymbol{fq}^+, \boldsymbol{fq}^-\} - \boldsymbol{fq}^+ \leq \log\left(\exp(\boldsymbol{fq}^+) + \exp(\boldsymbol{fq}^-)\right) - \boldsymbol{fq}^+,$$

the inequality holds by the log-sum-exp bound. And the last term above can be more generally organized into our Exponential-MMD loss as follows:

$$\mathcal{L}_{EMMD} = -\mathbb{E}_{x_i \in \mathcal{B}_T} \log \frac{\exp(\boldsymbol{f}_i\boldsymbol{q}_i^+/\tau)}{\exp(\boldsymbol{f}_i\boldsymbol{q}_i^+/\tau) + \exp(\boldsymbol{f}_i\boldsymbol{q}_i^-/\tau)}. \tag{10}$$

where $\tau$ is the temperature hyper-parameter, note that we set the temperature the same as that in Eqn. 5 (i.e. $\tau = 0.05$). Thus the Exponential-MMD loss can be obtained by the calculated results in the memory bank without additional computing costs.

## 5 Experiments

### 5.1 Experimental Setup

**Datasets and benchmarks.** We conduct experiments on three benchmark datasets: **Office-Home** [31] contains 65 classes from four distinct domains (Real, Clipart, Art, Product) and a total of 15,500 images. **VisDA-2017** [32] is a large-scale dataset, with 12-class synthetic-to-real object recognition tasks. The dataset consists of 152k synthetic images from the source domain while 55k real object images from the target domain. **DomainNet** [33] is originally a large-scale multi-source domain adaptation benchmark, which contains 6 domains with 345 classes. Similar to the setting of [34, 35],

we select four domains (Real, Clipart, Painting and Sketch) with 126 classes as the single-source unsupervised domain adaptation benchmark and construct seven single-source adaptation scenarios from the selected four domains. All results are the average of three random runs.

**Implementation Details.** We adopt the Resnet-50 [36] as backbone for Office-Home and ResNet-101 for VisDA, and conduct ResNet-34 for DomainNet similar to previous work [34, 35]. We use the same basic experimental setting in [1, 2] for a fair comparison. For the network architecture, the feature extractor consists of the backbone and a full-connected layer with batch normalization, and the classification head is full-connected with weight normalization. Following SHOT, the source model is supervised by smooth labels, the source model initializes the network, and we only train the feature extractor. The learning rate for the backbone is 10 times greater than that of the additional layer. The learning rate for the backbone is set as 2e-2 on Office-Home, 5e-4 on VisDA, and 1e-2 on DomainNet. We train 30 epochs for Office-Home, 60 epochs for VisDA, and 30 epochs for DomainNet. More training details are delineated in Appendix B.

**Baselines.** We compare *DaC* with multiple source-present and source-free domain adaptation baselines. Here we briefly introduce some of the most related state-of-the-art source-free methods: SHOT [1] and CPGA [3] exploit pseudo label prediction to achieve class-wise adaptation, NRC [2] and G-SFDA [5] strengthen the local structure by neighborhood information from source model, and SHOT++ [37] is a two-stage extension of SHOT, which adds the rotation prediction auxiliary task [38] to SHOT [1] in the first stage and trains the second stage in a semi-supervised manner (MixMatch [39]). *SF* in tables is short for source-free.

## 5.2 Comparison with State-of-the-arts

We compare *DaC* with the state-of-the-art methods on the Office-Home, VisDA, and DomainNet. As the results for **VisDA** shown in Table 2, our method surpasses all baselines in terms of average accuracy, including the recent source-present method BCDM and the most recent source-free method CPGA (87.3% *v.s.* 86.0%). For the **DomainNet**, Table 3 illustrates the proposed *DaC* has significantly outperformed the best source-free baseline by more than 3 percent (68.3 v.s. 65.1), and outperforms all source-present and source-free methods *w.r.t* average accuracy. Similar observation on the results of **Office-Home** can be found in Table 4. The reported results indicate the superiority of our method.

We further compare the effectiveness of our framework with the state-of-the-art SFUDA methods, in which CPGA [3] focuses on class-level supervision while NRC [2] focuses on neighborhood consistency. We show the UMAP visualization results in Fig. 1, and the accuracy curve in Fig. 3. Since the early accuracy of CPGA is low, Fig. 3 records the curve from 18 epochs. While CPGA

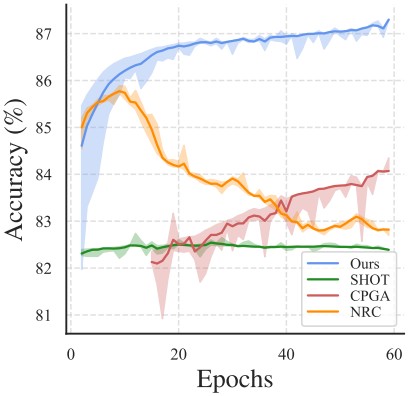

Figure 3: Average accuracy curve on VisDA over 60 epochs.

suffers false prediction within each cluster, NRC source bias local clusters due to the loss of global context, causing target error amplification. Our method can achieve fast convergence speed and stable high classification performance thanks to the effectiveness of the proposed divide and contrast scheme.

To validate the scalability of our model, we extend DaC into a two-stage version called DaC++. The second training stage is the same as SHOT++ for a fair comparison. Table 1 shows that DaC++ outperforms SHOT++ by more than 1 percent, and DaC++ is quite close to the target supervised learning performance even without any target annotations.

Table 1: Comparison of DaC++ and other state-of-the-art methods on VisDA.

| VisDA | NRC [2] | CPGA [3] | SHOT++ [37] | DaC | DaC++ | target-supervised |
|---|---|---|---|---|---|---|
| Avg. (%) | 85.9 | 86.0 | 87.3 | 87.3 | 88.6 | 89.6 |

Table 2: Accuracy (%) on the VisDA dataset (ResNet-101).

| Method | SF | plane | bicycle | bus | car | horse | knife | mcycl | person | plant | sktbrd | train | truck | Avg. |
|---|---|---|---|---|---|---|---|---|---|---|---|---|---|---|
| ResNet-101 [36] | ✗ | 55.1 | 53.3 | 61.9 | 59.1 | 80.6 | 17.9 | 79.7 | 31.2 | 81.0 | 26.5 | 73.5 | 8.5 | 52.4 |
| CDAN [40] | ✗ | 85.2 | 66.9 | 83.0 | 50.8 | 84.2 | 74.9 | 88.1 | 74.5 | 83.4 | 76.0 | 81.9 | 38.0 | 73.9 |
| SWD [41] | ✗ | 90.8 | 82.5 | 81.7 | 70.5 | 91.7 | 69.5 | 86.3 | 77.5 | 87.4 | 63.6 | 85.6 | 29.2 | 76.4 |
| MCC [42] | ✗ | 88.7 | 80.3 | 80.5 | 71.5 | 90.1 | 93.2 | 85.0 | 71.6 | 89.4 | 73.8 | 85.0 | 36.9 | 78.8 |
| STAR [43] | ✗ | 95.0 | 84.0 | 84.6 | 73.0 | 91.6 | 91.8 | 85.9 | 78.4 | 94.4 | 84.7 | 87.0 | 42.2 | 82.7 |
| BCDM [44] | ✗ | 95.1 | 87.6 | 81.2 | 73.2 | 92.7 | 95.4 | 86.9 | 82.5 | 95.1 | 84.8 | 88.1 | 39.5 | 83.4 |
| 3C-GAN [4] | ✓ | 94.8 | 73.4 | 68.8 | 74.8 | 93.1 | 95.4 | 88.6 | 84.7 | 89.1 | 84.7 | 83.5 | 48.1 | 81.6 |
| SHOT [1] | ✓ | 94.3 | 88.5 | 80.1 | 57.3 | 93.1 | 94.9 | 80.7 | 80.3 | 91.5 | 89.1 | 86.3 | 58.2 | 82.9 |
| G-SFDA [5] | ✓ | 96.1 | 88.3 | 85.5 | 74.1 | 97.1 | 95.4 | 89.5 | 79.4 | 95.4 | 92.9 | 89.1 | 42.6 | 85.4 |
| NRC [2] | ✓ | **96.8** | **91.3** | 82.4 | 62.4 | 96.2 | 95.9 | 86.1 | 80.6 | 94.8 | 94.1 | 90.4 | 59.7 | 85.9 |
| CPGA [3] | ✓ | 95.6 | 89.0 | 75.4 | 64.9 | 91.7 | **97.5** | 89.7 | 83.8 | 93.9 | 93.4 | 87.7 | **69.0** | 86.0 |
| **DaC** | ✓ | 96.6 | 86.8 | **86.4** | **78.4** | **96.4** | 96.2 | **93.6** | 83.8 | **96.8** | **95.1** | **89.6** | 50.0 | **87.3** |

Table 3: Accuracy (%) on the DomainNet dataset (ResNet-34). The * baselines are implemented by the official codes.

| Method | SF | Rw→Cl | Rw→Pt | Pt→Cl | Cl→Sk | Sk→Pt | Rw→Sk | Pt→Rw | Avg. |
|---|---|---|---|---|---|---|---|---|---|
| ResNet-34 [36] | ✗ | 58.4 | 62.5 | 56.0 | 50.1 | 41.9 | 48.2 | 70.1 | 56.7 |
| MME [34] | ✗ | 70.0 | 67.7 | 69.0 | 56.3 | 64.8 | 61.0 | 76.1 | 66.4 |
| CDAN [40] | ✗ | 65.0 | 64.9 | 63.7 | 53.1 | 63.4 | 54.5 | 73.2 | 62.5 |
| VDA* [42] | ✗ | 63.5 | 65.7 | 62.6 | 52.7 | 53.6 | 62.0 | 74.9 | 62.1 |
| GVB* [45] | ✗ | 68.2 | **69.0** | 63.2 | 56.6 | 63.1 | **62.2** | 73.8 | 65.2 |
| BAIT* [46] | ✓ | 64.7 | 65.4 | 62.1 | 57.1 | 61.8 | 56.7 | 73.2 | 63 |
| SHOT* [1] | ✓ | 67.1 | 65.1 | 67.2 | 60.4 | 63 | 56.3 | 76.4 | 65.1 |
| G-SFDA* [5] | ✓ | 63.4 | 67.5 | 62.5 | 55.3 | 60.8 | 58.3 | 75.2 | 63.3 |
| NRC* [2] | ✓ | 67.5 | 68.0 | 67.8 | 57.6 | 59.3 | 58.7 | 74.3 | 64.7 |
| **DaC** | ✓ | **70.0** | 68.8 | **70.9** | **62.4** | **66.8** | 60.3 | **78.6** | **68.3** |

Table 4: Accuracy (%) on the Office-Home dataset (ResNet-50).

| Method | SF | Ar→Cl | Ar→Pr | Ar→Rw | Cl→Ar | Cl→Pr | Cl→Rw | Pr→Ar | Pr→Cl | Pr→Rw | Rw→Ar | Rw→Cl | Rw→Pr | Avg. |
|---|---|---|---|---|---|---|---|---|---|---|---|---|---|---|
| ResNet-50 [36] | ✗ | 34.9 | 50.0 | 58.0 | 37.4 | 41.9 | 46.2 | 38.5 | 31.2 | 60.4 | 53.9 | 41.2 | 59.9 | 46.1 |
| MCD [13] | ✗ | 48.9 | 68.3 | 74.6 | 61.3 | 67.6 | 68.8 | 57.0 | 47.1 | 75.1 | 69.1 | 52.2 | 79.6 | 64.1 |
| CDAN [40] | ✗ | 50.7 | 70.6 | 76.0 | 57.6 | 70.0 | 70.0 | 57.4 | 50.9 | 77.3 | 70.9 | 56.7 | 81.6 | 65.8 |
| BNM [47] | ✗ | 52.3 | 73.9 | 80.0 | 63.3 | 72.9 | 74.9 | 61.7 | 49.5 | 79.7 | 70.5 | 53.6 | 82.2 | 67.9 |
| BDG [48] | ✗ | 51.5 | 73.4 | 78.7 | 65.3 | 71.5 | 73.7 | 65.1 | 49.7 | 81.1 | **74.6** | 55.1 | 84.8 | 68.7 |
| SHOT [1] | ✓ | 56.9 | 78.1 | 81.0 | 67.9 | 78.4 | 78.1 | 67.0 | 54.6 | 81.8 | 73.4 | 58.1 | 84.5 | 71.6 |
| G-SFDA [5] | ✓ | 57.9 | 78.6 | 81.0 | 66.7 | 77.2 | 77.2 | 65.6 | 56.0 | 82.2 | 72.0 | 57.8 | 83.4 | 71.3 |
| CPGA [3] | ✓ | **59.3** | 78.1 | 79.8 | 65.4 | 75.5 | 76.4 | 65.7 | **58.0** | 81.0 | 72.0 | **64.4** | 83.3 | 71.6 |
| NRC [2] | ✓ | 57.7 | **80.3** | **82.0** | 68.1 | **79.8** | 78.6 | 65.3 | 56.4 | **83.0** | 71.0 | 58.6 | **85.6** | 72.2 |
| **DaC** | ✓ | 59.1 | 79.5 | 81.2 | **69.3** | 78.9 | **79.2** | 67.4 | 56.4 | 82.4 | 74.0 | 61.4 | 84.4 | **72.8** |

## 5.3 Ablation Analysis

**Role of *Divide and Contrast* paradigm**. To study the effectiveness of our proposed *Divide and Contrast* paradigm, we conduct contrastive learning without splitting target data. We strengthen the local or class-level global structure with two contrastive learning schemes. *Scheme-S* focuses on obtaining discriminative class-wise clusters using a contrastive learning method similar to CPGA [3]. Specifically, the positive prototype is the class centroid $w.r.t.$ the anchor's pseudo label, while other centroids are the negative samples. *Scheme-T* only enhances the local structure by instance discrimination learning following [22, 20], the positive prototype is the anchor's corresponding feature in the memory bank, and the negative samples are the remaining features in the memory bank.

As shown in Table 5, both schemes improve the performance of self-training thanks to the contrastive framework. However, *Scheme-S* is vulnerable to the noisy pseudo labels while the *Scheme-T* can not further transfer the robust class-level classification into target outliers. Our method, in comparison, outperforms all the alternatives by a large margin.

**Component-wise Analysis**. we conduct our ablation study by isolating each part of our method on VisDA. As the results shown in Table 6, each component of our methods helps to enhance the performance, in which the adaptive contrastive learning framework makes the most contributions to

Table 5: Comparison with different contrastive learning schemes in the best accuracy over 60 epochs on VisDA.

| SHOT | Scheme-S | Scheme-T | DaC | Avg. (%) |
|:---:|:---:|:---:|:---:|:---:|
| ✓ | | | | 82.9 |
| ✓ | ✓ | | | 84.1 |
| ✓ | | ✓ | | 84.4 |
| | | | ✓ | **87.3** |

the promotion of accuracy (over 3% points). Besides, both strong augmentation and local structure enhance the local neighborhood information for each sample, which benefit the theoretical consistency $\mathcal{R}_{\mathcal{D}_T}(h)$ and improve the performance. Last but not least, removing the distribution alignment degrades the average accuracy to 86.5%, which means both our linear and exponential memory-bank-based MMD are helpful. The $\mathcal{L}_{EMMD}$ outperforms $\mathcal{L}_{LMMD}$ because the non-negative exponential form is upper bound of the other, and is stable in batch training.

Table 6: Ablation study of different losses (**left**) and different modules (**right**) on VisDA.

| BackBone | $\mathcal{L}_{self}$ | $\mathcal{L}_{con}$ | $\mathcal{L}_{LMMD}$ | $\mathcal{L}_{EMMD}$ | Avg. |
|:---:|:---:|:---:|:---:|:---:|:---:|
| ✓ | | | | | 59.5 |
| ✓ | ✓ | | | | 83.3 |
| ✓ | ✓ | ✓ | | | 86.5 |
| ✓ | ✓ | ✓ | ✓ | | 87.0 |
| ✓ | ✓ | ✓ | | ✓ | **87.3** |

| Method | Acc |
|:---:|:---:|
| DaC | **87.3** |
| DaC w/o Local Structure | 86.7 |
| DaC w/o Strong Augmentation | 85.6 |

# 6  Conclusion

In this paper, we have presented Divide and Contrast (DaC), a novel learning paradigm for the SFUDA problem that can inherit the advantages of (global) class-wise and (local) neighbors consistency approaches while sidestepping their limitations. The key idea is to divide the target data according to the prediction confidence of the source hypothesis (source-like v.s. target-specific) and apply customized learning strategies that can best fit the data property. We achieve this goal via a proposed adaptive contrastive learning where different groups of data samples are learned under a unified framework. We also proposed a MMD loss based on a memory bank to transfer the knowledge from the source-like domain to the target-specific data. While promising performance has been achieved using our proposed approach, it would be an interesting future avenue to investigate how to extend the DaC framework to more DA tasks, e.g. semi-supervised SFUDA or source-free open-set DA.

# Acknowledgments

This work was supported in part by the Guangdong Basic and Applied Basic Research Foundation (No.2020B1515020048), in part by National Key Research and Development Program of China (2020AAA0107200) and NSFC(61876077), in part by Open Research Projects of Zhejiang Lab (No.2019KD0AD01/017), in part by the National Natural Science Foundation of China (No.61976250, No.U1811463), in part by the Guangzhou Science and technology project (No.202102020633).

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
