# Divide and Contrast: Source-free Domain Adaptation via Adaptive Contrastive Learning (Supplementary Material)

Ziyi Zhang[1], Weikai Chen[3], Hui Cheng[2], Zhen Li[4,5], Siyuan Li[6], Liang Lin[2], Guanbin Li[2†]

[1]National Key Laboratory of Novel Software Technology, Nanjing University, Nanjing, China
[2]School of Computer Science and Engineering, Sun Yat-sen University, Guangzhou, China
[3] Tencent America, [4] The Chinese University of Hong Kong, Shenzhen, China
[5] Shenzhen Research Institute of Big Data, Shenzhen, China
[6] AI Lab, School of Engineering, Westlake University, Hangzhou, China
zhangziyi@lamda.nju.edu.cn, liguanbin@mail.sysu.edu.cn

## A   Theoretical Details

Our *Divide and Contrast* paradigm mainly divides the target data $\mathcal{D}_T$ into source-like samples $\mathcal{D}_S$ and target-specific outliers $\mathcal{D}_O$ via the network predictions from the source classifier. We claim the consistency robustness (Claim A.1) of the source-like samples. We further show in Theorem A.2 a upper bound of task error on target domain, and design our objective function motivated by constraining the upper bound.

We first review the main notations and assumptions in Section 3. Considering a $C$-way classification task, our model consists of source classifier and feature extractor $\bar{h} = g_s \circ \phi$, which maps input space $\mathbb{R}^I$ to prediction vector space $\mathbb{R}^C$, and $h(x) = \arg\max_c \bar{h}(x)_{[c]}$. Slightly different from the theoretical derivation, we use $h = g_s \circ \phi$ to denote our model in Section 4 to simplify the symbolic notations. Following in [25, 26, 27, 28], we denote $\mathcal{D}_{T_c}$ as the conditional distribution (probability measure) of $\mathcal{D}_T$ given the ground truth $y = c$, and also assume that the supports of $\mathcal{D}_{T_i}$ and $\mathcal{D}_{T_j}$ are disjoint for all $i \neq j$. This canonical assumption shows that the target distribution consists of $C$ class-wise subpopulations, and different class subpopulations have disjoint supports.

In order to study the local structure, we define that the suitable set of input transformations $\mathcal{B}(x) \subset \mathcal{D}_T$ takes the general form:

$$\mathcal{B}(x) = \{x' : \exists A \in \mathcal{A} \ s.t. \ ||(x' - A(x)|| < r\},$$

where $||\cdot||$ is $L_1$ distance function, $r > 0$ is a small radius, and $\mathcal{A}$ is a set of data augmentations. $\mathcal{B}(x)$ generally depicts the local structures around sample $x$, which can be understand as the neighborhood set in [2, 5] and the set of augmented data in [37, 49]. To this end, we further introduce the population consistency error on $\mathcal{D}_T$ to demonstrate the consistency robustness of predictions from the model within $\mathcal{B}(x)$:

$$\mathcal{R}_{\mathcal{D}_T}(h) = \mathbb{E}_{x \sim \mathcal{D}_T}[\mathbb{1}(\exists x' \in \mathcal{B}(x) \ s.t. \ h(x') \neq h(x))].$$

Following [25, 27, 26], we study target domain relies on the *expansion property*, which implies the continuity of data distributions in each class-wise subpopulations. For this, we first define the neighborhood function of a sample $x \in \mathcal{D}_{T_i}$ as: $\mathcal{N}(x) := \mathcal{D}_{T_i} \cap \{x'|\mathcal{B}(x) \cap \mathcal{B}(x') \neq \emptyset\}$, as well as that of a set $S \subset \mathcal{D}_T$ as: $\mathcal{N}(S) := \cup_{x \in S} \mathcal{N}(x)$.

---

[*]Corresponding author.

[†]Corresponding author.

36th Conference on Neural Information Processing Systems (NeurIPS 2022).

**Definition A.1** (($q, \gamma$)**-constant expansion** [26])**.** *We say $Q$ satisfies $(q, \gamma)$-constant expansion for some constant $q, \gamma \in (0, 1)$, if for any set $S \subset Q$ with $\mathbb{P}_Q[S] > q$, we have $\mathbb{P}_Q[\mathcal{N}(S) \setminus S] > \min\{\gamma, \mathbb{P}_Q[S]\}$.*

Our *Divide and Contrast* paradigm selects confident samples (from target data) with prediction probability greater than a threshold $\tau$, and regard them as source-like samples:

$$\mathcal{D}_S = \{x_i | \max_c \bar{h}(x_i) \geq \tau, x_i \in \mathcal{D}_T\},$$

and the rest target data is target-specific samples $\mathcal{D}_O = \mathcal{D}_T \setminus \mathcal{D}_S$. Define the conditional distribution of source-like samples as $\mathcal{D}_{S_i} = \mathcal{D}_S \cap \mathcal{D}_{T_i}$. The definition is similar for $\mathcal{D}_{O_i}$. The following claim guarantees the consistency and robustness of source-like samples:

**Claim A.1.** *Suppose $h$ is $L_h$-Lipschitz w.r.t the $L_1$ distance, there exists threshold $\tau \in (0, 1)$ such that the source-like set $\mathcal{D}_S$ is consistency robust, i.e. $\mathcal{R}_{\mathcal{D}_S}(h) = 0$. More specifically,*

$$\tau \geq \frac{L_h r}{4} + \frac{1}{2}.$$

*Proof of Claim A.1.* Suppose $\mathcal{D}_S$ is defined by $\tau \geq \frac{L_h r}{4} + \frac{1}{2}$. If $\exists\ x, x' \in \mathcal{D}_S$, $x' \in \mathcal{B}(x) \cap \mathcal{D}_S$ s.t. $\bar{h}(x) \neq \bar{h}(x')$. Denote $h(x) = i$, $h(x') = j$, since $i \neq j$, there is $\bar{h}(x)_{[i]} - \bar{h}(x')_{[j]} \geq 2\tau - 1$, and $\bar{h}(x')_{[j]} - \bar{h}(x)_{[i]} \geq 2\tau - 1$. Then we have:

$$L_h \|x - x'\| \geq \|\bar{h}(x) - \bar{h}(x')\| \geq |\bar{h}(x)_{[i]} - \bar{h}(x')_{[j]}| + |\bar{h}(x)_{[j]} - \bar{h}(x')_{[i]}| \geq 4\tau - 2 \geq L_h r,$$

Since $x' \in \mathcal{B}(x)$, $\|x - x'\| < r$, and Lipschitz constant $L_h > 0$, this forms a contradiction with $L_h \|x - x'\| \geq L_h r$. Thus, $\forall x \in \mathcal{D}_S, x' \in \mathcal{B}(x) \cap \mathcal{D}_S$, the network predictions are consistent, i.e. $\mathcal{R}_{\mathcal{D}_S}(h) = 0$. $\qquad\square$

Assume we have a pseudo-labeler $h_{pl}$ based on the source model. In Theorem A.2, we establish a upper bound of target error based on expansion property.

**Theorem A.2.** *Suppose the condition of Claim 3.1 holds and $\mathcal{D}_T, \mathcal{D}_S$ satisfies $(q, \gamma)$-constant expansion. Then the expected error of model $h \in \mathcal{H}$ on target domain $\mathcal{D}_T$ is bounded,*

$$\epsilon_{\mathcal{D}_T}(h) \leq (\mathbb{P}_{\mathcal{D}_T}[h(x) \neq h_{pl}(x)] - \epsilon_{\mathcal{D}_S}(h_{pl}) + q) \frac{\mathcal{R}_{\mathcal{D}_T}(h)(1 + \gamma)}{\gamma \cdot \min\{q, \gamma\}} + \max_{i \in [C]}\{d_{\mathcal{H}\Delta\mathcal{H}}(\mathcal{D}_{S_i}, \mathcal{D}_{O_i})\} + \lambda,$$
$$(11)$$

We have proved that source-like samples are consistency robust (Claim A.1). For the source-like samples, we further assume that $\epsilon_{\mathcal{D}_S}(h) < \epsilon_{\mathcal{D}_T}(h)$ and $\mathbb{P}_{x \sim \mathcal{D}_S}[h(x) \neq h_{pl}(x)] < \mathbb{P}_{x \sim \mathcal{D}_T}[h(x) \neq h_{pl}(x)]$, which empirically holds since all source-like samples with confident predictions [18]. To prove Theorem A.2, we first use the expansion property to study the error bound of $\mathcal{D}_S$, and introduce new notations as follows. Let $\mathcal{M}^i(h) = \{x : h(x) \neq i, x \in \mathcal{D}_{S_i}\}$ denote the source-like samples where the model makes mistakes. The definition is similar for $\mathcal{M}(h_{pl})$, the source-like samples where the pseudolabeler makes mistakes. We introduces three disjoint subsets of $\mathcal{M}^i(h)$ following [26]: $\mathcal{M}^i_1 = \{x : h(x) = h_{pl}(x), h(x) \neq i\}$, $\mathcal{M}^i_2 = \{x : h(x) \neq h_{pl}(x), h(x) \neq i, h_{pl}(x) \neq i\}$, $\mathcal{M}^i_3 = \{x : h(x) \neq h_{pl}(x), h(x) \neq i, h_{pl}(x) = i\}$. $\mathcal{M}^i_1 \cup \mathcal{M}^i_2 \subseteq \mathcal{M}^i(h_{pl}) \cap \mathcal{M}^i(h)$, where both $h$ and $h_{pl}$ makes mistakes.

When $h$ fits the pseudolabels well, i.e. $\mathbb{P}_{x \sim \mathcal{D}_{S_i}}[\mathbb{1}(h(x) \neq h_{pl}(x))] - \epsilon_{\mathcal{D}_{S_i}}(h_{pl}) \leq \gamma$, the Lemma A.3 in [26] states that:
$$\mathbb{P}_{\mathcal{D}_{S_i}}[\mathcal{M}^i_1 \cup \mathcal{M}^i_2] = \mathbb{P}_{\mathcal{D}_{S_i}}[\mathcal{M}^i_1] + \mathbb{P}_{\mathcal{D}_{S_i}}[\mathcal{M}^i_2] \leq q \qquad (12)$$

Thus we define $I = \{i \in [C] | \mathbb{P}_{x \sim \mathcal{D}_{S_i}}[\mathbb{1}(h(x) \neq h_{pl}(x))] - \epsilon_{\mathcal{D}_{S_i}}(h_{pl}) \leq \gamma\}$, where $[C] = \{1, 2, \ldots, C\}$. The following lemma bounds the probability of $[C] \setminus I$.

**Lemma A.1** (Upper bound on the subpopulations of $[C] \setminus I$)**.** *Under the setting of Theorem A.2,*

$$\sum_{i \in [C] \setminus I} \mathbb{P}_{\mathcal{D}_S}[\mathcal{D}_{S_i}] \leq \frac{1}{\gamma}(\mathbb{P}_{x \in \mathcal{D}_S}[h(x) \neq h_{pl}(x)] - \epsilon_{\mathcal{D}_S}(h_{pl}) + q)$$

*Proof of Lemma A.1.* For any $i \in [C]$, the disjoint three parts $\mathcal{M}_2^i, \mathcal{M}_3^i$, and $(\mathcal{M}^i(h_{pl}) \cap \overline{\mathcal{M}^i(h)})$ have inconsistent predictions between $h$ and $h_{pl}$. Thus, $\mathcal{M}_2^i \cup \mathcal{M}_3^i \cup (\mathcal{M}^i(h_{pl}) \cap \overline{\mathcal{M}^i(h)}) \subseteq \{x : h(x) \neq h_{pl}(x), x \in \mathcal{D}_{S_i}\}$, and we have:

$$\mathbb{P}_{\mathcal{D}_{S_i}}[\mathcal{M}_2^i] + \mathbb{P}_{\mathcal{D}_{S_i}}[\mathcal{M}_3^i] + \mathbb{P}_{\mathcal{D}_{S_i}}[\mathcal{M}^i(h_{pl}) \cap \overline{\mathcal{M}^i(h)}] \leq \mathbb{P}_{x \sim \mathcal{D}_{S_i}}[h(x) \neq h_{pl}(x)] \quad (13)$$

It is not hard to verify that $\mathcal{M}^i(h_{pl}) \setminus \mathcal{M}_1^i \subseteq \mathcal{M}_2^i \cup (\mathcal{M}^i(h_{pl}) \cap \overline{\mathcal{M}^i(h)})$, then:

$$\begin{aligned}
\epsilon_{\mathcal{D}_{S_i}}(h_{pl}) - \mathbb{P}_{\mathcal{D}_{S_i}}[\mathcal{M}_1^i] &\leq \mathbb{P}_{\mathcal{D}_{S_i}}[\mathcal{M}^i(h_{pl}) \setminus \mathcal{M}_1^i] \\
&\leq \mathbb{P}_{\mathcal{D}_{S_i}}[\mathcal{M}_2^i] + \mathbb{P}_{\mathcal{D}_{S_i}}[\mathcal{M}^i(h_{pl}) \cap \overline{\mathcal{M}^i(h)}] \\
\text{by Eqn. } 13 &\leq \mathbb{P}_{x \sim \mathcal{D}_{S_i}}[h(x) \neq h_{pl}(x)] - \mathbb{P}_{\mathcal{D}_{S_i}}[\mathcal{M}_3^i]
\end{aligned} \quad (14)$$

Then, we can write:

$$\begin{aligned}
\mathbb{P}_{x \sim \mathcal{D}_S}[h(x) \neq h_{pl}(x)] &= \sum_{i \in I} \mathbb{P}_{x \sim \mathcal{D}_{S_i}}[h(x) \neq h_{pl}(x)]\mathbb{P}_{\mathcal{D}_S}[\mathcal{D}_{S_i}] \\
&\quad + \sum_{i \in [C] \setminus I} \mathbb{P}_{x \sim \mathcal{D}_{S_i}}[h(x) \neq h_{pl}(x)]\mathbb{P}_{\mathcal{D}_S}[\mathcal{D}_{S_i}] \\
\text{by Eqn. } 14 &\geq \sum_{i \in I} \mathbb{P}_{\mathcal{D}_S}[\mathcal{D}_{S_i}](\epsilon_{\mathcal{D}_{S_i}}(h_{pl}) - \mathbb{P}_{\mathcal{D}_{S_i}}[\mathcal{M}_1^i]) \\
&\quad + \sum_{i \in [C] \setminus I} \mathbb{P}_{x \sim \mathcal{D}_{S_i}}[h(x) \neq h_{pl}(x)]\mathbb{P}_{\mathcal{D}_S}[\mathcal{D}_{S_i}] \\
\text{by Eqn. } 12 \text{ and Definition of } I &> \sum_{i \in I} \mathbb{P}_{\mathcal{D}_S}[\mathcal{D}_{S_i}](\epsilon_{\mathcal{D}_{S_i}}(h_{pl}) - q) \\
&\quad + \sum_{i \in [C] \setminus I} (\epsilon_{\mathcal{D}_{S_i}}(h_{pl}) + \gamma)\mathbb{P}_{\mathcal{D}_S}[\mathcal{D}_{S_i}] \\
&\geq \epsilon_{\mathcal{D}_{S_i}}(h_{pl}) - q + \gamma \sum_{i \in [C] \setminus I} \mathbb{P}_{\mathcal{D}_S}[\mathcal{D}_{S_i}]
\end{aligned} \quad (15)$$

By organizing the Eqn. 15, we complete the proof. $\qquad \square$

**Lemma A.2.** *Under the condiction of Theorem A.2, for any $i \in I$, the task error on $\mathcal{D}_{S_i}$ is bounded by:*

$$\epsilon_{\mathcal{D}_{S_i}}(h) \leq \mathbb{P}_{x \sim \mathcal{D}_{S_i}}[h(x) \neq h_{pl}(x)] - \epsilon_{\mathcal{D}_{S_i}}(h_{pl}) + 2q$$

*Proof of Lemma A.2.* For $i \in I$, we can write:

$$\begin{aligned}
\epsilon_{\mathcal{D}_{S_i}}(h) &= \mathbb{P}_{\mathcal{D}_{S_i}}[\mathcal{M}_2^i] + \mathbb{P}_{\mathcal{D}_{S_i}}[\mathcal{M}_3^i] + \mathbb{P}_{\mathcal{D}_{S_i}}[\mathcal{M}_1^i] \\
\text{by Eqn. } 14 &\leq \mathbb{P}_{\mathcal{D}_{S_i}}[\mathcal{M}_2^i] + 2\mathbb{P}_{\mathcal{D}_{S_i}}[\mathcal{M}_1^i] + \mathbb{P}_{x \sim \mathcal{D}_{S_i}}[h(x) \neq h_{pl}(x)] - \epsilon_{\mathcal{D}_{S_i}}(h_{pl}) \\
&\leq 2(\mathbb{P}_{\mathcal{D}_{S_i}}[\mathcal{M}_2^i] + \mathbb{P}_{\mathcal{D}_{S_i}}[\mathcal{M}_1^i]) + \mathbb{P}_{x \sim \mathcal{D}_{S_i}}[h(x) \neq h_{pl}(x)] - \epsilon_{\mathcal{D}_{S_i}}(h_{pl}) \\
\text{by Eqn. } 12 &\leq 2q + \mathbb{P}_{x \sim \mathcal{D}_{S_i}}[h(x) \neq h_{pl}(x)] - \epsilon_{\mathcal{D}_{S_i}}(h_{pl})
\end{aligned} \quad (16)$$

$\qquad \square$

Based on the results above, we turn to bound the target error on the whole target data.

*Proof of Theorem A.2.*

$$\epsilon_{\mathcal{D}_T}(h) = \sum_{i=1}^{C} \mathbb{P}_{\mathcal{D}_T}[\mathcal{D}_{T_i}]\epsilon_{\mathcal{D}_{T_i}}(h) = \sum_{i \in G_1} \mathbb{P}_{\mathcal{D}_T}[\mathcal{D}_{T_i}]\epsilon_{\mathcal{D}_{T_i}}(h) + \sum_{i \in G_2} \mathbb{P}_{\mathcal{D}_T}[\mathcal{D}_{T_i}]\epsilon_{\mathcal{D}_{T_i}}(h) \quad (17)$$

Following [25], we define $G_1 = \{i \in [C] : \mathcal{R}_{\mathcal{D}_{T_i}}(h) \leq \min\{q, \gamma\}\}, G_2 = \{i \in [C] : \mathcal{R}_{\mathcal{D}_{T_i}}(h) > \min\{q, \gamma\}\}$, by the Lemma 2 in [25], we also claim that, for $\forall i \in G_1$:

$$\epsilon_{\mathcal{D}_{T_i}}(h) \leq q, \quad (18)$$

otherwise, by the expansion property A.1, $\mathbb{P}_{\mathcal{D}_{T_i}}[\mathcal{N}(\mathcal{M}^i(h))\backslash\mathcal{M}^i(h)] > \min\{q,\gamma\}$. We claim that the $\mathcal{N}(\mathcal{M}^i(h)) \setminus \mathcal{M}^i(h)$ is subset of $\mathcal{R}_{\mathcal{D}_{T_i}}(h)$. If not, $\exists\, x \in \mathcal{N}(\mathcal{M}^i(h)) \setminus \mathcal{M}^i(h)$, say $x \in \mathcal{N}(\mathcal{M}^i(h) \setminus \mathcal{R}_{\mathcal{D}_{T_i}}(h))$, by the definition of neighborhood, $\exists\, x' \in \mathcal{M}^i(h) \setminus \mathcal{R}_{\mathcal{D}_{T_i}}(h)$ $s.t.$ $\exists\, x'' \in \mathcal{B}(x) \cap \mathcal{B}(x')$. By the definition of $\mathcal{R}_{\mathcal{D}_{T_i}}(h)$, we have $h(x) = h(x') = h(x'') = i$, which contradicts to $x' \in \mathcal{M}^i(h)$. Therefore, the consistency error on the subpopulations $\mathcal{R}_{\mathcal{D}_{T_i}}(h) \geq \mathbb{P}_{\mathcal{D}_{T_i}}[\mathcal{N}(\mathcal{M}(h)) \setminus \mathcal{M}(h)] > \min\{q,\gamma\}$, which contradicts to the definition of $G_1$ that $\mathcal{R}_{\mathcal{D}_{T_i}}(h) \leq \min\{q,\gamma\}$.

For $i \in G_2$, by the Lemma 1 in [25], we have:

$$\sum_{i \in G_2} \mathbb{P}_{\mathcal{D}_T}[\mathcal{D}_{T_i}] \leq \frac{\mathcal{R}_{\mathcal{D}_T}(h)}{\min\{q,\gamma\}}, \tag{19}$$

otherwise, $\mathcal{R}_{\mathcal{D}_T}(h) > \sum_{i \in G_2} \mathbb{P}_{\mathcal{D}_T}[\mathcal{D}_{T_i}] \min\{q,\gamma\} > \mathcal{R}_{\mathcal{D}_T}(h)$, which forms contradiction.

The consistency error on the subpopulation of $G_2$ is greater than $\min\{\gamma,q\}$, we use our divided source-like set to estimate the target error $\epsilon_{\mathcal{D}_{T_i}}$. Following the Theorem 2 in [28], for all $h$ in the model space of $C$-way classification task $\mathcal{H}$, we have:

$$\begin{aligned}
\epsilon_{\mathcal{D}_{T_i}}(h) &= \mathbb{P}_{\mathcal{D}_{T_i}}[\mathcal{D}_{S_i}]\epsilon_{\mathcal{D}_{S_i}}(h) + \mathbb{P}_{\mathcal{D}_{T_i}}[\mathcal{D}_{O_i}]\epsilon_{\mathcal{D}_{O_i}}(h) \\
&\leq \mathbb{P}_{\mathcal{D}_{T_i}}[\mathcal{D}_{S_i}]\epsilon_{\mathcal{D}_{S_i}}(h) + \mathbb{P}_{\mathcal{D}_{T_i}}[\mathcal{D}_{O_i}]\left(\epsilon_{\mathcal{D}_{S_i}}(h) + d_{\mathcal{H}\Delta\mathcal{H}}(\mathcal{D}_{S_i}, \mathcal{D}_{O_i}) + \lambda_i\right) \\
&\leq \epsilon_{\mathcal{D}_{S_i}}(h) + d_{\mathcal{H}\Delta\mathcal{H}}(\mathcal{D}_{S_i}, \mathcal{D}_{O_i}) + \lambda_i,
\end{aligned} \tag{20}$$

where $\lambda_i = \min_{h \in \mathcal{H}}\{\epsilon_{\mathcal{D}_{S_i}}(h) + \epsilon_{\mathcal{D}_{O_i}}(h)\}$. Organizing Eqn. 18,19,20 into Eqn. 17, we have:

$$\begin{aligned}
\epsilon_{\mathcal{D}_T} &\leq \sum_{i \in G_1} \mathbb{P}_{\mathcal{D}_T}[\mathcal{D}_{T_i}]q + \sum_{i \in G_2} \mathbb{P}_{\mathcal{D}_T}[\mathcal{D}_{T_i}]\epsilon_{\mathcal{D}_{S_i}}(h) + \max_i\{d_{\mathcal{H}\Delta\mathcal{H}}(\mathcal{D}_{S_i}, \mathcal{D}_{O_i})\} + \lambda' \\
&\leq q + \sum_{i \in G_2} \mathbb{P}_{\mathcal{D}_T}[\mathcal{D}_{T_i}]\epsilon_{\mathcal{D}_{S_i}}(h) + \max_i\{d_{\mathcal{H}\Delta\mathcal{H}}(\mathcal{D}_{S_i}, \mathcal{D}_{O_i})\} + \lambda' \\
\text{(by Eqn. 19)} &\leq \frac{\mathcal{R}_{\mathcal{D}_T}(h)}{\min\{q,\gamma\}}\epsilon_{\mathcal{D}_S}(h) + \max_i\{d_{\mathcal{H}\Delta\mathcal{H}}(\mathcal{D}_{S_i}, \mathcal{D}_{O_i})\} + \lambda' + q
\end{aligned} \tag{21}$$

where $\lambda' = \min_{h \in \mathcal{H}}\{\epsilon_{\mathcal{D}_S}(h) + \epsilon_{\mathcal{D}_O}(h)\}$. Then, there is:

$$\begin{aligned}
\epsilon_{\mathcal{D}_S}(h) &= \sum_{i \in I} \epsilon_{\mathcal{D}_{S_i}}(h)\mathbb{P}_{\mathcal{D}_S}[\mathcal{D}_{S_i}] + \sum_{i \in [c]\setminus I} \mathbb{P}_{\mathcal{D}_S}[\mathcal{D}_{S_i}]\epsilon_{\mathcal{D}_{S_i}}(h) \\
\text{(by Lemma A.1)} &\leq \frac{1}{\gamma}(\mathbb{P}_{x\sim\mathcal{D}_S}[h(x) \neq h_{pl}(x)] - \epsilon_{\mathcal{D}_S}(h_{pl}) + q) + \sum_{i \in [c]\setminus I} \mathbb{P}_{\mathcal{D}_S}[\mathcal{D}_{S_i}]\epsilon_{\mathcal{D}_{S_i}}(h) \\
\text{(by Lemma A.2)} &\leq \frac{1}{\gamma}(\mathbb{P}_{x\sim\mathcal{D}_S}[h(x) \neq h_{pl}(x)] - \epsilon_{\mathcal{D}_S}(h_{pl}) + q) \\
&\quad + \mathbb{P}_{x\in\mathcal{D}_S}[h(x) \neq h_{pl}(x)] - \epsilon_{\mathcal{D}_S}(h_{pl}) + 2q \\
&\leq \frac{1+\gamma}{\gamma}(\mathbb{P}_{x\sim\mathcal{D}_T}[h(x) \neq h_{pl}(x)] - \epsilon_{\mathcal{D}_S}(h_{pl}) + q) + q.
\end{aligned} \tag{22}$$

Combining the results of Eqn. 21 and Eqn. 22, we prove the result of Theorem A.2. Specifically, in Eqn. 11 , the $\lambda = \lambda' + q(1 + \frac{\mathcal{R}_{\mathcal{D}_T}(h)(1+\gamma)}{\gamma \cdot \min\{q,\gamma\}})$ is a constant $w.r.t$ the expansion constant $q$ and task risk of ideal optimal model. $\qquad\square$

# B  Additional Experimental Details

## B.1  Implementation Details

We train our model on four Nvidia Geforce GTX 1080Ti graphic cards, using SGD with a momentum of 0.9, and a weight decay of 0.0005. We conduct experiments on VisDA, Office-Home and DomainNet and set the batch size to 64 for all benchmarks. The initial learning rate is set as 5e-4

for VisDA, 2e-2 for Office-Home, and 1e-2 for DomainNet. The total epoch is set as 60 for VisDA, 30 for Office-Home and DomainNet. We apply the learning rate scheduler $\eta = \eta_0(1 + 15p)^{3/4}$ following [1], where training process $p$ changes from 0 to 1, and we further reduce the learning rate by a factor of 10 after 40 epochs on Visda, 15 epochs on Office-Home and DomainNet. We find that most hyperparameters of *DaC* do not require to be heavily tuned. As can be seen in Table 7, the performance is not sensitive to the choice of $\tau_c$, and we set the confidence threshold $\tau_c$ as 0.95 for all experiments following [50, 35]. We adopt a set of hyperparameters $\alpha = 0.5, \beta = 0.5, K = 5$ for the large scale benchmarks VisDA and DomainNet, and $\alpha = 0.7, \beta = 0.3, K = 3$ for most transfer senarios of Office-Home.

Table 7: Sensitive analysis of $\tau_c$.

| $\tau_c$ | 0.91 | 0.93 | 0.95 | 0.97 | 0.98 | target-supervised |
|---|---|---|---|---|---|---|
| Avg. (%) | 87.06 | 87.27 | 87.34 | 87.39 | 87.19 | 89.6 |

## B.2 Baseline Methods on DomainNet

We compare *DaC* with source-present and source-free domain adaptation methods. DomainNet is widely used in multi-source domain adaptation tasks, and its subset is used as one of the benchmarks of single-source domain adaptation benchmark by [34]. The results of MME [34] and CDAN [40] are copied from [34]. The rest source-present and source-free methods are implemented by their official codes. We choose the learning rate for all baselines by five-fold cross-validation, and apply the training scheduler of their own.

## C   Algorithm for DaC

As shown in Algorithm 1, our method consists of self-training by pseudo-labeling, adaptive contrastive learning, and distribution alignment. After self-training to achieve preliminary class-wise adaptation, we divide target data as source-like and target-specific to conduct representation learning. The adaptive contrastive learning framework exploits local and global information and improves feature discriminability. Distribution alignment reduces the mismatching between source-like and target-specific samples.

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

**Algorithm 1** Training of DaC

---

**Require:** unlabeled target data $\mathcal{D}_T = \{x_i\}_{i=1}^{n_t}$, source model:$g_s, \phi_s$, augmentation set $\mathcal{A} = \{\mathcal{A}_w, \mathcal{A}_s\}$, threshold $\tau$, batch size $B$.
    **Initialization:** target model $h = g \circ \phi, g = g_s, \phi = \phi_s$, memory bank by forward computation: $\mathcal{F} = \phi_s(\mathcal{D}_T)$, source-like and target specific features by Eqn. 6.
    **while** $e <$ MaxEpoch **do**
        Obtain the pseudo labels $\tilde{\mathcal{Y}}$ based on Eqn. 2.
        **for** $t = 1 \rightarrow$ NumIters **do**
            From $\mathcal{D}_T \times \tilde{\mathcal{Y}}$, draw a mini-batch $\mathcal{B}_t = \{(x_i, \tilde{y}_i), i \in \{1, 2, \ldots, B\}\}$.    ▷ batch-training
            **for** $b = 1 \rightarrow B$ **do**
                $p_i^w = \delta(h(\mathcal{A}_w(x_i))), p_i^s = \delta(h(\mathcal{A}_s(x_i)))$;
                $\boldsymbol{f}_i = \phi(\mathcal{A}_w(x_i)), \boldsymbol{f}_i^s = \phi(\mathcal{A}_s(x_i))$;
                $\boldsymbol{z}_i = m\boldsymbol{z}_i + (1-m)\boldsymbol{f}_i$;                ▷ update memory bank
                update the source-like and target-specific sample based on Eqn. 4.    ▷ divide
            **end for**
            Compute $\mathcal{L}_{self}$ using by Eqn. 3                ▷ self-training;
            Generate prototypes $\boldsymbol{k}^+, \{\boldsymbol{w}\}, \{\boldsymbol{v}\}$ from memory bank;
            Compute $\mathcal{L}_{con}$ by Eqn. 5;                ▷ contrast
            Compute $\mathcal{L}_{EMMD}$ by Eqn. 10;          ▷ distribution alignment
            $\mathcal{L} = \mathcal{L}_{\text{con}} + \alpha\mathcal{L}_{\text{self}} + \beta\mathcal{L}_{EMMD}$;
            $\phi = SGD(\mathcal{L}, \phi)$.
        **end for**
    **end while**
    **Output** $g_s \circ \phi$

---