# OpenReview forum: "Divide and Contrast: Source-free Domain Adaptation via Adaptive Contrastive Learning"
_NeurIPS.cc/2022/Conference — NeurIPS 2022 Accept_

### Official Review · Reviewer_shxH · 2022-06-30

**Rating:** 8
**Confidence:** 3
**Soundness:** 3 good
**Presentation:** 3 good
**Contribution:** 3 good

**Summary:**

This paper is about unsupervised domain adaptation when the source data is unavailable at the time of adaptation. Authors term this problem as source-free unsupervised domain adaptation (SFUDA). Authors make the insightful  observation that existing SFUDA techniques use all pseudo labels on target data including noisy labels or enforce local feature consistency at the expense of being source-biased. Motivated by these observations, authors propose a new SFUDA approach that they call Divide and Contrast (DaC) which divides the target data into two disjoint groups: source-like samples and target-specific samples. An adaptive learning framework is presented that treats each of these target sample groups differently during the training process. Authors present both theoretical results and numerical results that illustrate the superiority of DaC over state of the art methods for SFUDA.

**Questions:**

1. Line 51: Provide a brief explanation for the "memory bank".
2. Line 260: What is SHOT? Cite a reference for that.


**Limitations:**

Authors do not explicitly discuss the limitations of the proposed method, but mention semi-supervised SFUDA  and source-free open-set DA as possible research extension topics. If space permits, it may be interesting to provide a few more sentences about each of these topics and how DaC might fare in those cases.

**Strengths And Weaknesses:**

Strengths
Authors illustrate well (e.g., Fig. 1) the tradeoffs involved in the global and local approaches to SFUDA.

The proposed DaC approach for SFUDA is a novel approach to address the limitations of prior methods to SFUDA.

Authors present theoretical derivations to justify the DaC approach.

Numerical results shown on multiple datasets are pretty convincing of the superiority of DaC approach.

Weaknesses
"Memory bank" is invoked early on without sufficient explanation. It is explained in Section 4.2.2, but may make the paper more readable if this introduction could be provide earlier in the manuscript.

No error bars are provided for the numerical results.

---

> ### Author Response · Authors · 2022-08-02
> **Response to Reviewer shxH**
>
> Thank you for appreciating our work! Specific concerns are addressed below.
>
> > Line 51: Provide a brief explanation for the "memory bank".
>
> Considering your constructive suggestion and the coherence of the reading, we will add the following brief description of the memory bank in Line 47.
>
> ‘’…with memory bank. Specifically, memory bank consists of representations of all target samples, and momentum updated in the training stage [3][15][16]. Thanks to the high…’’
>
> >Line 260: What is SHOT? Cite a reference for that.
>
> SHOT [1], which leverages self-training to achieve class-wise adaptation (as described in the Abstract), is one of our most related baselines. We will include the corresponding reference in the revision.
>
> >No error bars are provided for the numerical results.
>
> We conducted our method three times and found error bars are relatively small. Taking the average accuracy on VisDA as an example, the mean results of multiple runs is 87.3, while the deviation between the best and worst runs is 0.027. Thus, we follow all of our baselines [1][2][3][5] which do not provide an error bar in numerical results. We will include it in the revision.
>
> >Future work discussion.
>
> The semi-supervised SFUDA can obtain a few labels for each category, and the labeled samples can assist DaC to generate more robust class-wise prototypes. How to make better use of annotation information under our proposed DaC framework is an interesting open question in this setting.
>
> The open-set DA setting contains some out-of-distribution samples, which may lead to negative transfer that is detrimental to the performance. Under the framework of DaC, these OOD samples could be treated as target-specific samples and get better exploited in our contrastive learning pipeline to achieve more discriminative local features. This could be another interesting future avenue of extending our work.

---

> > ### Comment · Reviewer_shxH · 2022-08-08
> > **Reviewer shxH reply**
> >
> > Authors have addressed the error bars issue that I have raised. Other reviewers have raised some concerns about comparisons with SOTA approaches, but it appears that the authors have adequately addressed most of the concerns.

---

### Official Review · Reviewer_rqR4 · 2022-07-10

**Rating:** 5
**Confidence:** 5
**Soundness:** 3 good
**Presentation:** 4 excellent
**Contribution:** 2 fair

**Summary:**

This paper proposes a new source-free unsupervised domain adaptation method named DaC. The key idea is to leverage the advantages of existing “global” methods and “local” counterparts. Specifically, DaC uses the source model to split the target data into source-like and target-specific samples. After that, an adaptive contrastive learning strategy is used to achieve class-wise adaptation (global) and local consistency (local). Finally, MMD is used to minimize the distribution mismatch between source-like samples and target-specific samples. The proposed method achieves the best performance on widely used benchmarks.


**Questions:**

1. Line 262: change “trains” to “train”
2. Any reason for missing the experiments on Digit datasets and the Office31 dataset?


**Limitations:**

Not applicable.


**Strengths And Weaknesses:**

---

Originality: the proposed method is a combination of well-known techniques, but achieves good performance. Integrating contrastive learning into source-free domain adaptation is novel and brings new insights into this community.

---

Quality:

Strengths: this work is technically sound with theoretical proof and empirical evaluation. The effectiveness of the proposed regularizations are evaluated on various downstream tasks. The work is complete.

Weakness: (1) the major concern is the performance. As shown in Table 1 and Table 3, compared to existing SOTA methods (NRC and CPGA), the performance improvement is quite limited. Although it outperforms the baseline SHOT by a large margin, it is still hard to convince readers. (2) The proposed method is not evaluated on Digit datasets (MNIST, SVHN, and USPS) and the Office31 dataset which are two important benchmarks. Any reason behind this?

---

Clarity: this paper is well written and well organized. It is easy to follow. Detailed implementation details are also provided in the supplementary.

---

Significance: compared to the baseline SHOT, the result is important. Specifically, the proposed method brings a larger improvement than SHOT. However, it is not clear to me whether other methods can borrow the same idea and get performance improvement.

---

---

> ### Author Response · Authors · 2022-08-02
> **Response to Reviewer rqR4**
>
> Thank you for the thorough review. The two main concerns are addressed below.
>
> > Compared to existing SOTA methods (NRC and CPGA), the performance improvement is quite limited.
> >
>
> We have **significantly outperformed the best source-free baseline (i.e. SHOT) on DomainNet**, the most difficult benchmark that we have experimented with in this paper, by **more than 3 percent** (68.3 v.s. 65.1).  In fact, **our approach can achieve a bigger leap in the performance boost in a more difficult and larger dataset with a greater domain gap**. Intuitively, we divide the target samples into source-like and target-specific ones by the source classifier. If the domain gap is small, most of the target samples would be regarded as source-like in the perspective of the source model. In this case, most of the domain adaptation methods can easily exploit their advantages and achieve good classification accuracy, leaving little room for improvement. If the domain gap is large, most of the existing methods would fail to generate accurate pseudo-labeling results. In contrast, our divide-and-contrast framework can take full advantage of both the global and local structures of target data via data segmentation and customized learning strategies for data subsets. Hence, we are able to significantly improve the performance in difficult and noisy settings.
>
> Office-Home is a relatively small benchmark. DaC can still achieve more than 1.2 points average accuracy improvement (over 12 transfer scenarios) compared with most of our baselines.
>
> For the VisDA dataset, we have achieved more than 1 percent advantage in average accuracy over the best source-free baseline approach (87.3 v.s. 86.0). The numerical improvement is less significant than that of DomainNet as the performance gain in VisDA is close to saturation. We provide more numerical comparisons between DaC, DaC++(extension of our model), CPGA [3], NRC [2], and the oracle (target supervised) results as follows:
>
> | VisDA | NRC[2] | CPGA [3] | DaC | DaC++ | target-supervised |
> | :-: | :-: | :-: | :-: | :-: | :-: |
> | Average Acc | 85.9 | 86.0 | 87.3 | 88.6 | 89.6 |
>
> Note that “target-supervised” represents that all target samples are supervised with ground-truth labels, and this oracle result is copied from [a]. DaC++ is our two-stage extension by the same second stage training with [a]. Both DaC and DaC++ can achieve better accuracy that is much closer to the target-supervised result than [2][3]. Hence, we believe the improvement on the benchmarks is substantial.
>
> Besides the numerical results, we offer the **accuracy curve comparison** on VisDA in Figure 3. It shows that DaC can achieve **more** **stable training, faster convergence speed, and better classification performance than the other candidate approaches**.
>
> [a] Liang, Jian, et al. "Source data-absent unsupervised domain adaptation through hypothesis transfer and labeling transfer." In TPAMI, 2021.
>
> > The proposed method is not evaluated on Digit datasets (MNIST, SVHN, and USPS) and the Office31 dataset which are two important benchmarks. Any reason behind this?
> >
>
> As for Digit datasets, we do not evaluate DaC on them since they are relatively simple benchmarks. The average result of SHOT [1] is 98.3, which is quite close to the target supervised result 98.4. In addition, other most recent baselines [2][3][5] are not tested in Digit datasets.
>
> Due to the limited space, we only chose the more challenging Office-Home as representative of the Office series datasets (Office31, Office-Home, and Office-Caltech). For more thorough comparisons, we add the experimental results on  Office31 as follows. In Office31, our method surpasses both source-free and source-available baselines.
>
> | Office31 | Source free | AD | AW | DA | DW | WA | WD | Average |
> | :-: | :-: | :-: | :-: | :-: | :-: | :-: | :-: | :-: |
> | GVB [40] | No | 95 | 94.8 | 76.8 | 98.7 | 73.7 | 100.0 | 89.3 |
> | SHOT [1] | Yes | 94 | 90.1 | 74.7 | 98.4 | 74.3 | 99.9 | 88.6 |
> | DaC (Ours) | Yes | 94.2 | 91.7 | 76.8 | 98.1 | 75.7 | 99.8 | 89.4 |

---

### Official Review · Reviewer_4tiB · 2022-07-11

**Rating:** 5
**Confidence:** 4
**Soundness:** 3 good
**Presentation:** 3 good
**Contribution:** 3 good

**Summary:**

This paper presents a source-free domain adaptation method. Previous methods either use self-supervised pseudo labeling to conduct class-wise global alignment or leverage local structure to enforce feature consistency. This work combines the idea of both. The proposed method divides target samples into source-like and target-specific ones. Source-like samples are used for global class clustering and target-specific samples are used for learning local structures. The two are further aligned using maximum mean discrepancy loss.

**Questions:**

See weaknesses above.

**Strengths And Weaknesses:**

Strengths are as follows.

The idea is interesting. The target samples are divided based on confidence output of source classifier. Different groups are treated differently, either globally in class-level or locally in instance-level. Two different groups are aligned to encourage consistency that is also interesting. The presentation is generally good. Ablation study is conducted for each part.

Weaknesses are as follows.

The major weakness is the performance compared to prior work. It shows that the results of the proposed method are just marginally good compared to [2] and [3] in 2 out of 3 datasets (table 1 and table 3).

The performance is good in table 2. Can the authors explain more about the baseline results? For example, how the official code was used and how the authors ensure the hyper parameters are reasonably well tuned?

---

> ### Author Response · Authors · 2022-08-02
> **Response to Reviewer 4tiB**
>
> We thank reviewer 4tiB for the thorough review. The two main concerns are addressed below.
>
> > It shows that the results of the proposed method are just marginally good compared to [2] and [3] in 2 out of 3 datasets (table 1 and table 3).
> >
>
> We have **significantly outperformed the best source-free baseline (i.e. SHOT) on DomainNet**, the most difficult benchmark that we have experimented with in this paper, by **more than 3 percent** (68.3 v.s. 65.1).  In fact, **our approach can achieve a bigger leap in the performance boost in a more difficult and larger dataset with a greater domain gap**. Intuitively, we divide the target samples into source-like and target-specific ones by the source classifier. If the domain gap is small, most of the target samples would be regarded as source-like in the perspective of the source model. In this case, most of the domain adaptation methods can easily exploit their advantages and achieve good classification accuracy, leaving little room for improvement. If the domain gap is large, most of the existing methods would fail to generate accurate pseudo-labeling results. In contrast, our divide-and-contrast framework can take full advantage of both the global and local structures of target data via data segmentation and customized learning strategies for data subsets. Hence, we are able to significantly improve the performance in difficult and noisy settings.
>
> Office-Home is a relatively small benchmark. DaC can still achieve more than 1.2 points average accuracy improvement (over 12 transfer scenarios) compared with most of our baselines.
>
> For the VisDA dataset, we have achieved more than 1 percent advantage in average accuracy over the best source-free baseline approach (87.3 v.s. 86.0). The numerical improvement is less significant than that of DomainNet as the performance gain in VisDA is close to saturation. We provide more numerical comparisons between DaC, DaC++(extension of our model), CPGA [3], NRC [2], and the oracle (target supervised) results as follows:
>
> | VisDA | NRC[2] | CPGA [3] | DaC | DaC++ | target-supervised |
> | :-: | :-: | :-: | :-: | :-: | :-: |
> | Average Acc | 85.9 | 86.0 | 87.3 | 88.6 | 89.6 |
>
> Note that “target-supervised” represents that all target samples are supervised with ground-truth labels, and this oracle result is copied from [a]. DaC++ is our two-stage extension by the same second stage training with [a]. Both DaC and DaC++ can achieve better accuracy that is much closer to the target-supervised result than [2][3]. Hence, we believe the improvement on the benchmarks is substantial.
>
> Besides the numerical results, we offer the **accuracy curve comparison** on VisDA in Figure 3. It shows that DaC can achieve **more** **stable training, faster convergence speed, and better classification performance than the other candidate approaches.**.
>
> > The performance is good in table 2. Can the authors explain more about the baseline results? For example, how the official code was used and how the authors ensure the hyperparameters are reasonably well tuned?
> >
>
> As the discussion above, the performance gain of DaC is more significant for the more challenging DomainNet dataset.
>
> In terms of implementation details, to ensure fair comparisons with all SFUDA baselines, we first trained the source model by supervised learning, and then conduct model adaptation on the target domain using the same batch size, learning rate, and training epochs as that of our approach. For the source-present domain adaptation baselines, we copy the results of MME and CDAN from the original papers and implement VDA and GVB with the same learning rate and training epochs. We directly use the hyper-parameters provided in their released codes.
>
>
> [a] Liang, Jian, et al. "Source data-absent unsupervised domain adaptation through hypothesis transfer and labeling transfer." In TPAMI, 2021.

---

### Official Review · Reviewer_xRX9 · 2022-07-11

**Rating:** 5
**Confidence:** 4
**Soundness:** 3 good
**Presentation:** 3 good
**Contribution:** 3 good

**Summary:**

The authors proposed a method for source-free unsupervised domain adaptation (SFUDA) task. The key idea is to combine the advantages of global alignment [1] and feature consistency [2]. The authors divided the target data into source-like and target-specific samples and
treat them by different learning methods. The authors demonstrate extensive experiments on three datasets and verify the performance of the proposed method

**Questions:**

1. When do the method update the source-like set and class centroids, after one batch or one epoch？

**Limitations:**

Yes

**Strengths And Weaknesses:**

Strengths
1.The paper is well-organized and easy to follow.
2. The "divide and contrast" strategy is simple but effective; it can exploit both the global and local structures of target data.
3. The proposed Exponential-MMD loss is novel, it also makes sense to align the source-like and target-specific samples to reduce distribution mismatch.
4. The experiment and ablations are sufficient to support the conclusion.

Weaknesses
1. The section of preliminaries and analysis is a bit unclear to introduce the task.
2. A more recent work that outperforms proposed approach ([a]) is not compared against.
[a] Liang, Jian, et al. "Source data-absent unsupervised domain adaptation through hypothesis transfer and labeling transfer." IEEE Transactions on Pattern Analysis and Machine Intelligence (2021).
3. Eqn. 4 and Eqn. 5 share the same parameter \tau but they are different from each other. The threshold \tau in Eqn. 4 is set to 0.95 according to supplementary material following [47], however, I could not find the reference [47]. Besides, the choice of the threshold \tau is not included in the ablation study, it would be necessary to see the ablation of choosing the threshold \tau.
4. As described in Discussion(L200-L207), unlike previous method [3], Eqn. 5 jointly achieves class-wise adaptation and instance-wise adaptation. It would be interesting if the authors could compare the proposed loss with separate class-wise adaptation and instance-wise adaptation losses.
5. When do the method update the source-like set and class centroids, after one batch or one epoch.

---

> ### Author Response · Authors · 2022-08-02
> **Response to Reviewer xRX9 (2/2)**
>
> > Eqn. 4 and Eqn. 5 share the same parameter $\tau$ but they are different from each other. The threshold $\tau$ in Eqn. 4 is set to 0.95 according to the supplementary material following [47], however, I could not find the reference [47]. Besides, the choice of the threshold $\tau$ is not included in the ablation study, it would be necessary to see the ablation of choosing the threshold $\tau$.
> >
>
> We first apologize for the abused use of $\tau$ (in Eqn. 4 and Eqn. 5), and the missing reference [47] due to the submission error of supplementary material. We will include all missing references in the revised version. We add the ablation study of $\tau$ on VisDA, and the results are shown in the following table.
>
> | $\tau$ | Avg acc |
> | :-: | :-: |
> | 0.91 | 87.06 |
> | 0.93 | 87.27 |
> | 0.95 | 87.34 |
> | 0.97 | 87.39 |
> | 0.98 | 87.19 |
>
> As can be seen, the performance is not sensitive to the choice of $\tau$.
>
> [44] Shai Ben-David, et al. "A theory of learning from different domains." In Machine learning, 2010.
>
> [45] Jian Liang, et al. "Source data-absent unsupervised domain adaptation through hypothesis transfer and labeling transfer." In TPAMI, 2021.
>
> [46] Qizhe Xie, et al. "Unsupervised data augmentation for consistency training." In NeurIPS, 2020.
>
> [47] Alex Kurakin, et al. "Fixmatch: Simplifying semi-supervised learning with consistency and confidence." In NeurIPS, 2020
>
>
> > Unlike previous method [3], Eqn. 5 jointly achieves class-wise adaptation and instance-wise adaptation. It would be interesting if the authors could compare the proposed loss with separate class-wise adaptation and instance-wise adaptation losses.
> >
>
> Eqn. 5 is the proposed contrastive loss that enhances both global and local structures. We have already included the suggested comparison **in our ablation analysis** (Section 5.3. Role of *Divide and Contrast* paradigm), in which *Scheme-S* only achieves class-wise adaptation like method [3] while *Scheme-T* regards all samples as target-specific and only conducts instance-wise adaptation. The results in Table 4 demonstrate the effectiveness of our framework.
>
> > The section of preliminaries and analysis is a bit unclear to introduce the task.
> >
>
> We agree that Section 3 is theory-heavy and a bit difficult to follow. We intended to provide the theoretical insight behind our proposed method in Section 3. In particular, *Claim 3.1* shows that the source-like set is consistency-robust under a sufficiently large threshold of prediction probability. This lays the foundation of our data segmentation strategy. *Theorem 3.2* states that the target risk is bounded by three parts that are later formulated as the three losses of our method. In the revision, we will polish the writing to make it more informative and better connected to our task.
>
>
> > When do the method update the source-like set and class centroids, after one batch or one epoch?
> >
>
> We update the source-like set and class centroids **after one batch**. (see Supplementary Material C. Algorithm 1. the line commented with “divide”)

---

> > ### Comment · Reviewer_xRX9 · 2022-08-08
> > **Reviewer xRX9 Reply**
> >
> > Thanks for the reply. My concerns are addressed.

---

> ### Author Response · Authors · 2022-08-02
> **Response to Reviewer xRX9 (1/2)**
>
> We thank reviewer xRX9 for the detailed review! We reply point-by-point here.
>
> > A more recent work that outperforms proposed approach ([a]) is not compared against.
> >
>
> SHOT++ [a] is a two-stage extension of SHOT [1]. After adding the rotation prediction auxiliary task [b] to SHOT in the first stage, the second stage of [a] is trained in a semi-supervised manner (MixMatch [c]). Since we aim to propose a simple and scalable paradigm for source-free unsupervised domain adaptation, we believe it is a bit unfair to compare our end-to-end framework with the two-stage extension of [1]. For a more thorough comparison, we first compare our method with the result of the end-to-end version of [a] (denoted as SHOT+ here). We then add the second stage training to compare with a full version of [a].
>
> | VisDA | Avg acc |
> | :-: | :-: |
> | SHOT+ [a] | 85.5 |
> | DaC++ | 87.3 |
> | Target Supervised | 89.6 |
>
> | Office-Home | Avg | AC | AP | AR | CA | CP | CR | PA | PC | PR | RA | RC | RP |
> | :-: | :-: | :-: | :-: | :-: | :-: | :-: | :-: | :-: | :-: | :-: | :-: | :-: | :-: |
> | SHOT+ [a] | 72.0 | 57.7 | 79.1 | 81.5 | 67.6 | 77.9 | 77.8 | 68.1 | 55.8 | 82 | 72.8 | 59.7 | 84.4 |
> | DaC (Ours) | 72.8 | 59.1  | 79.5  | 81.2  | 69.3  | 78.9  | 79.2  | 67.4  | 56.4  | 82.4 | 74.0  | 61.4  | 84.4 |
>
> | DomainNet | Avg | Rw→Cl  | Rw→Pt  | Pt→Cl  | Cl→Sk | Sk→Pt  | Rw→Sk  | Pt→Rw |
> | :-: | :-: | :-: | :-: | :-: | :-: | :-: | :-: | :-: |
> | SHOT+ [a] | 66.4 | 67.7 | 65.6 | 69.3 | 62.1 | 64.9 | 57.6 | 77.7 |
> | DaC (Ours) | 68.3 | 70.0  | 68.8  | 70.9  | 62.4 |  66.8  | 60.3  | 78.6 |
>
> From the results above, even without the rotation prediction technique, DaC is comparable with SHOT+. After extending DaC to a two-stage version DaC++, we make another comparison with the full version of SHOT++ on the VisDA dataset.
>
> | VisDA | Avg Acc |
> | :-: | :-: |
> | SHOT++ [a] | 87.3 |
> | DaC++ | 88.6 |
> | Target Supervised | 89.6 |
>
> Note that “target-supervised” represents that all the target samples are supervised with ground-truth labels, and this oracle result is copied from [a]. As seen from the table, DaC++ outperforms SHOT++ by more than 1 percent in terms of the average accuracy while achieving performance very close to the oracle result. This validates the superiority and scalability of our framework.
>
> [a] Liang, Jian, et al. "Source data-absent unsupervised domain adaptation through hypothesis transfer and labeling transfer." In TPAMI, 2021.
>
> [b] S. Gidaris, et al, “Unsupervised representation learning by predicting image rotations,” in ICLR, 2018.
>
> [c] D. Berthelot, et al, “Mixmatch: A holistic approach to semisupervised learning,” in NeurIPS, 2019.

---

### Meta-Review · Area_Chair_DetB · 2022-08-27

**Recommendation:** Accept
**Confidence:** Certain

**Metareview:**

This paper proposes a relatively complicated method for source-free unsupervised domain adaptation, which integrates several techniques into a divide and contrast framework. The idea of dividing the target data into source-like subset and target-specific subset and employing global alignment and feature consistency for each subset is novel when the source data is inaccessible. The contrastive learning and memory-based MMD are novel in the context of source-free domain adaptation and introduce theoretical benefits in terms of the expansion theory and domain alignment theory, respectively. Reviewers were on the positive side while holding some concerns on the marginal improvement over the SoTA methods, which were addressed in the author rebuttal. AC generally agreed that the paper has introduced a novel and solid contribution to the field, with a nice connection between algorithmic methods and theoretical insights, and  recommended the paper for acceptance. Authors are suggested to incorporate all rebuttal material in the revision and if possible, to work out a recipe for easing the adoption of their relatively complicated framework that comes with many modules and loss terms.

**Award:**

No

---

### Decision · Program_Chairs · 2022-09-14

Accept